# MODELING THE DATA-GENERATING PROCESS IS NECESSARY FOR OUT-OF-DISTRIBUTION GENERALIZATION

**Jivat Neet Kaur, Emre Kıcıman, Amit Sharma**
Microsoft Research {t-kaurjivat,emrek,amshar}@microsoft.com

## ABSTRACT

Recent empirical studies on domain generalization (DG) have shown that DG algorithms that perform well on some distribution shifts fail on others, and no state-of-the-art DG algorithm performs consistently well on all shifts. Moreover, real-world data often has multiple distribution shifts over different attributes; hence we introduce *multi*-attribute distribution shift datasets and find that the accuracy of existing DG algorithms falls even further. To explain these results, we provide a formal characterization of generalization under multi-attribute shifts using a canonical causal graph. Based on the relationship between spurious attributes and the classification label, we obtain *realizations* of the canonical causal graph that characterize common distribution shifts and show that each shift entails different independence constraints over observed variables. As a result, we prove that any algorithm based on a single, fixed constraint cannot work well across all shifts, providing theoretical evidence for mixed empirical results on DG algorithms. Based on this insight, we develop *Causally Adaptive Constraint Minimization (CACM)*, an algorithm that uses knowledge about the data-generating process to *adaptively* identify and apply the correct independence constraints for regularization. Results on fully synthetic, MNIST, small NORB, and Waterbirds datasets, covering binary and multi-valued attributes and labels, show that adaptive dataset-dependent constraints lead to the highest accuracy on unseen domains whereas incorrect constraints fail to do so. Our results demonstrate the importance of modeling the causal relationships inherent in the data-generating process.

## 1 INTRODUCTION

To perform reliably in real world settings, machine learning models must be robust to distribution shifts – where the training distribution differs from the test distribution. Given data from multiple domains that share a common optimal predictor, the *domain generalization (DG)* task (Wang et al., 2021; Zhou et al., 2021) encapsulates this challenge by evaluating accuracy on an unseen domain. Recent empirical studies of DG algorithms (Wiles et al., 2022; Ye et al., 2022) have characterized different kinds of distribution shifts across domains. Using MNIST as an example, a *diversity* shift is when domains are created either by adding new values of a spurious attribute like rotation (e.g., Rotated-MNIST dataset (Ghifary et al., 2015; Piratla et al., 2020)) whereas a *correlation* shift is when domains exhibit different values of correlation between the class label and a spurious attribute like color (e.g., Colored-MNIST (Arjovsky et al., 2019)). Partly because advances in representation learning for DG (Ahuja et al., 2021; Krueger et al., 2021; Mahajan et al., 2021; Arjovsky et al., 2019; Li et al., 2018a; Sun & Saenko, 2016) have focused on either one of the shifts, these studies find that performance of state-of-the-art DG algorithms are not consistent across different shifts: algorithms performing well on datasets with one kind of shift fail on datasets with another kind of shift.

In this paper, we pose a harder, more realistic question: What if a dataset exhibits two or more kinds of shifts *simultaneously*? Such shifts over multiple attributes (where an attribute refers to a spurious high-level variable like rotation) are often observed in real data. For example, satellite imagery data demonstrates distribution shifts over time as well as the region captured (Koh et al., 2021). To study this question, we introduce *multi*-attribute distribution shift datasets. For instance, in our Col+Rot-MNIST dataset (see Figure 1), both the color and rotation angle of digits can shift

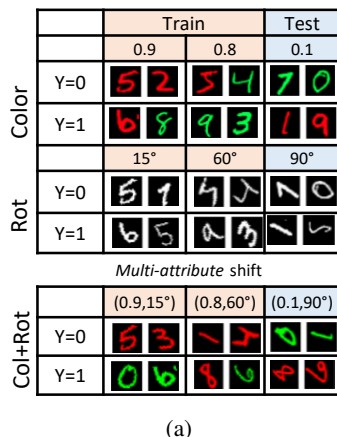

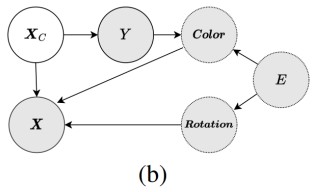

(b)

| Algo. | Color | Rotation | Col+Rot |
|---|---|---|---|
| ERM | $30.9 \pm 1.6$ | $61.9 \pm 0.5$ | $25.2 \pm 1.3$ |
| IRM | $50.0 \pm 0.1$ | $61.2 \pm 0.3$ | $39.6 \pm 6.7$ |
| MMD | $29.7 \pm 1.8$ | $62.2 \pm 0.5$ | $24.1 \pm 0.6$ |
| C-MMD | $29.4 \pm 0.2$ | $62.3 \pm 0.4$ | $32.2 \pm 7.0$ |
| *CACM* | $\mathbf{70.4 \pm 0.5}$ | $\mathbf{62.4 \pm 0.4}$ | $\mathbf{54.1 \pm 1.3}$ |

(a)      (c)

Figure 1: (a) Our *multi-attribute* distribution shift dataset Col+Rot-MNIST. We combine Colored MNIST (Arjovsky et al., 2019) and Rotated MNIST (Ghifary et al., 2015) to introduce distinct shifts over *Color* and *Rotation* attributes. (b) The causal graph representing the data generating process for Col+Rot-MNIST. *Color* has a correlation with $Y$ which changes across environments while *Rotation* varies independently. (c) Comparison with DG algorithms optimizing for different constraints shows the superiority of *Causally Adaptive Constraint Minimization (CACM)* (full table in Section 5).

across data distributions. We find that existing DG algorithms that are often targeted for a specific shift fail to generalize in such settings: best accuracy falls from 50-62% for individual shift MNIST datasets to <50% (lower than a random guess) for the multi-attribute shift dataset.

To explain such failures, we propose a causal framework for generalization under multi-attribute distribution shifts. We use a canonical causal graph to model commonly observed distribution shifts. Under this graph, we characterize a distribution shift by the type of relationship between spurious attributes and the classification label, leading to different *realized* causal DAGs. Using *d*-separation on the realized DAGs, we show that each shift entails distinct constraints over observed variables and prove that no conditional independence constraint is valid across all shifts. As a special case of *multi*-attribute, when datasets exhibit a *single*-attribute shift across domains, this result provides an explanation for the inconsistent performance of DG algorithms reported by Wiles et al. (2022); Ye et al. (2022). It implies that any algorithm based on a single, fixed independence constraint cannot work well across all shifts: there will be a dataset on which it will fail (Section 3.3).

We go on to ask if we can develop an algorithm that generalizes to different kinds of individual shifts as well as simultaneous *multi*-attribute shifts. For the common shifts modeled by the canonical graph, we show that identification of the correct regularization constraints requires knowing only the type of relationship between attributes and the label, not the full graph. As we discuss in Section 3.1, the type of shift for an attribute is often available or can be inferred for real-world datasets. Based on this, we propose *Causally Adaptive Constraint Minimization (CACM)*, an algorithm that leverages knowledge about the data-generating process (DGP) to identify and apply the correct independence constraints for regularization. Given a dataset with auxiliary attributes and their relationship with the target label, *CACM* constrains the model's representation to obey the conditional independence constraints satisfied by causal features of the label, generalizing past work on causality-based regularization (Mahajan et al., 2021; Veitch et al., 2021; Makar et al., 2022) to multi-attribute shifts.

We evaluate *CACM* on novel multi-attribute shift datasets based on MNIST, small NORB, and Waterbirds images. Across all datasets, applying the incorrect constraint, often through an existing DG algorithm, leads to significantly lower accuracy than the correct constraint. Further, *CACM* achieves substantially better accuracy than existing algorithms on datasets with multi-attribute shifts as well as individual shifts. Our contributions include:

- Theoretical result that an algorithm using a fixed independence constraint cannot yield an optimal classifier on all datasets.
- An algorithm, *Causally Adaptive Constraint Minimization (CACM)*, to adaptively derive the correct regularization constraint(s) based on the causal graph that outperforms existing DG algorithms.
- Multi-attribute shifts-based benchmarks for domain generalization where existing algorithms fail.

## 2 GENERALIZATION UNDER MULTI-ATTRIBUTE SHIFTS

We consider the supervised learning setup from (Wiles et al., 2022) where each row of train data $(\boldsymbol{x}_i, \boldsymbol{a}_i, y_i)_{i=1}^n$ contains input features $\boldsymbol{x}_i$ (e.g., X-ray pixels), a set of nuisance or spurious attributes $\boldsymbol{a}_i$ (e.g., vertical shift, hospital) and class label $y_i$ (e.g., disease diagnosis). The attributes represent variables that are often recorded or implicit in data collection procedures. Some attributes represent a property of the input (e.g., vertical shift) while others represent the domain from which the input was collected (e.g., hospital). The attributes affect the observed input features $\boldsymbol{x}_i$ but do not cause the target label, and hence are *spurious* attributes. The final classifier $g(\boldsymbol{x})$ is expected to use only the input features. However, as new values of attributes are introduced or as the correlation of attributes with the label changes, we obtain different conditional distributions $P(Y|\boldsymbol{X})$. Given a set of data distributions $\mathcal{P}$, we assume that the train data is sampled from distributions, $\mathcal{P}_{\mathcal{E}tr} = \{P_{E1}, P_{E2}, \cdots\} \subset \mathcal{P}$ while the test data is assumed to be sampled from a single unseen distribution, $\mathcal{P}_{\mathcal{E}te} = \{P_{Ete}\} \subset \mathcal{P}$. Attributes and class labels are assumed to be discrete.

### 2.1 RISK INVARIANT PREDICTOR FOR GENERALIZATION UNDER SHIFTS

The goal is to learn a classifier $g(\boldsymbol{x})$ using train domains such that it generalizes and achieves a similar, small risk on test data from unseen $P_{Ete}$ as it achieves on the train data. Formally, given a set of distributions $\mathcal{P}$, we define a risk-invariant predictor (Makar et al., 2022) as,

**Definition 2.1.** *Optimal Risk Invariant Predictor for $\mathcal{P}$ (from (Makar et al., 2022)) Define the risk of predictor $g$ on distribution $P \in \mathcal{P}$ as $R_P(g) = \mathbb{E}_{\boldsymbol{x}, y \sim P} \ell(g(\boldsymbol{x}), y)$ where $\ell$ is cross-entropy or another classification loss. Then, the set of risk-invariant predictors obtain the same risk across all distributions $P \in \mathcal{P}$, and set of the optimal risk-invariant predictors is defined as the risk-invariant predictors that obtain minimum risk on all distributions.*

$$g_{rinv} \in \arg \min_{g \in G_{rinv}} R_P(g) \; \forall P \in \mathcal{P} \; where \; G_{rinv} = \{g : R_P(g) = R_{P'}(g) \forall P, P' \in \mathcal{P}\} \quad (1)$$

An intuitive way to obtain a risk-invariant predictor is to consider only the parts of the input features ($\boldsymbol{X}$) that cause the label $Y$ and ignore any variation due to the spurious attributes. Let such latent, unobserved causal features be $\boldsymbol{X}_c$. Due to independence and stability of causal mechanisms (Peters et al., 2017), we can assume that $P(Y|\boldsymbol{X}_c)$ remains invariant across different distributions. Using the notion of risk invariance, we can now define the multi-attribute generalization problem as,

**Definition 2.2.** *Generalization under Multi-attribute shifts. Given a target label $Y$, input features $\boldsymbol{X}$, attributes $\boldsymbol{A}$, and latent causal features $\boldsymbol{X}_c$, consider a set of distributions $\mathcal{P}$ such that $P(Y|\boldsymbol{X}_c)$ remains invariant while $P(\boldsymbol{A}|Y)$ changes across individual distributions. Using a training dataset $(\boldsymbol{x}_i, \boldsymbol{a}_i, y_i)_{i=1}^n$ sampled from a subset of distributions $\mathcal{P}_{\mathcal{E}tr} \subset \mathcal{P}$, the generalization goal is to learn an optimal risk-invariant predictor over $\mathcal{P}$.*

**Special case of single-attribute shift.** When $|\boldsymbol{A}| = 1$, we obtain the single-attribute shift problem that is widely studied (Wiles et al., 2022; Ye et al., 2022; Gulrajani & Lopez-Paz, 2021).

### 2.2 A GENERAL PRINCIPLE FOR NECESSARY CONDITIONAL INDEPENDENCE CONSTRAINTS

In practice, the causal features $\boldsymbol{X}_c$ are unobserved and a key challenge is to learn $\boldsymbol{X}_c$ using the observed $(\boldsymbol{X}, Y, \boldsymbol{A})$. We focus on representation learning-based (Wang et al., 2021) DG algorithms, typically characterized by a regularization constraint that is added to a standard ERM loss such as cross-entropy. Table 1 shows three independence constraints that form the basis of many popular DG algorithms, assuming environment/domain $E$ as the attribute and $\ell$ as the main classifier loss. We now provide a general principle for deciding which constraints to choose for learning a risk-invariant predictor for a dataset.

Table 1: Statistic optimized by DG algorithms. match matches the statistic across $E$. $h$ is domain classifier (loss $\ell_d$) using shared representation $\phi$.

| Constraint | Statistic | Algo. |
|---|---|---|
| $\phi \perp\!\!\!\perp E$ | match $\mathbb{E}[\phi(x)|E] \; \forall \; E$ | MMD |
| | $\max_E \; \mathbb{E}[\ell_d(h(\phi(x)), E)]$ | DANN |
| | match $\mathrm{Cov}[\phi(x)|E] \; \forall \; E$ | CORAL |
| $Y \perp\!\!\!\perp E|\phi$ | match $\mathbb{E}[Y|\phi(x), E] \; \forall \; E$ | IRM |
| | match $\mathrm{Var}[\ell(g(x), y)|E] \; \forall \; E$ | VREx |
| $\phi \perp\!\!\!\perp E|Y$ | match $\mathbb{E}[\phi(x)|E, Y = y] \; \forall \; E$ | C-MMD |
| | $\max_E \; \mathbb{E}[\ell_d(h(\phi(x)), E)|Y = y]$ | CDANN |

We utilize a strategy from past work (Mahajan et al., 2021; Veitch et al., 2021) to use graph structure of the underlying data-generating process (DGP). We assume that the predictor can be represented as $g(\boldsymbol{x}) = g_1(\phi(\boldsymbol{x}))$ where $\phi$ is the representation. To learn a risk-invariant $\phi$, we identify the conditional independence constraints satisfied by causal features $\boldsymbol{X}_c$ in the causal graph and enforce that learnt representation $\phi$ should follow the same constraints. If $\phi$ satisfies the constraints, then any function $g_1(\phi)$ will also satisfy them. Below we show that the constraints are necessary under simple assumptions on the causal DAG representing the DGP for a dataset. All proofs are in Suppl. B.

**Theorem 2.1.** *Consider a causal DAG $\mathcal{G}$ over $\langle \boldsymbol{X}_c, \boldsymbol{X}, \boldsymbol{A}, Y \rangle$ and a corresponding generated dataset $(\boldsymbol{x}_i, \boldsymbol{a}_i, y_i)_{i=1}^n$, where $\boldsymbol{X}_c$ is unobserved. Assume that graph $\mathcal{G}$ has the following property: $\boldsymbol{X}_c$ is defined as the set of all parents of $Y$ ($\boldsymbol{X}_c \to Y$); and $\boldsymbol{X}_c, \boldsymbol{A}$ together cause $\boldsymbol{X}$ ($\boldsymbol{X}_c \to \boldsymbol{X}$, and $\boldsymbol{A} \to \boldsymbol{X}$). The graph may have any other edges (see, e.g., DAG in Figure 1(b)). Let $\mathcal{P}_\mathcal{G}$ be the set of distributions consistent with graph $\mathcal{G}$, obtained by changing $P(\boldsymbol{A}|Y)$ but not $P(Y|\boldsymbol{X}_c)$. Then the conditional independence constraints satisfied by $\boldsymbol{X}_c$ are necessary for a (cross-entropy) risk-invariant predictor over $\mathcal{P}_\mathcal{G}$. That is, if a predictor for $Y$ does not satisfy any of these constraints, then there exists a data distribution $P' \in \mathcal{P}_\mathcal{G}$ such that predictor's risk will be higher than its risk in other distributions.*

Thus, given a causal DAG, using $d$-separation on $\boldsymbol{X}_c$ and observed variables, we can derive the correct regularization constraints to be applied on $\phi$. This yields a general principle to learn a risk-invariant predictor. We use it to theoretically explain the inconsistent results of existing DG algorithms (Sec. 3.3) and to propose an Out-of-Distribution generalization algorithm *CACM* (Sec. 4). Note that constraints from *CACM* are necessary but not sufficient as $\boldsymbol{X}_c$ is not identifiable.

# 3 STUDYING DISTRIBUTION SHIFTS THROUGH A CANONICAL CAUSAL GRAPH

## 3.1 CANONICAL CAUSAL GRAPH FOR COMMON DISTRIBUTION SHIFTS

We consider a *canonical* causal graph (Figure 2) to specify the common data-generating processes that can lead to a multi-attribute shift dataset. Shaded nodes represent observed variables $\boldsymbol{X}, Y$; and the sets of attributes $\boldsymbol{A}_{\overline{ind}}$, $\boldsymbol{A}_{ind}$, and $E$ such that $\boldsymbol{A}_{\overline{ind}} \cup \boldsymbol{A}_{ind} \cup \{E\} = \boldsymbol{A}$. $\boldsymbol{A}_{\overline{ind}}$ represents the attributes correlated with label, $\boldsymbol{A}_{ind}$ the attributes that are independent of label, while $E$ is a special attribute for the domain/environment from which a data point was collected. Not all attributes need to be observed. For example, in some cases, only $E$ and a subset of $\boldsymbol{A}_{\overline{ind}}$, $\boldsymbol{A}_{ind}$ may be observed. In other cases, only $\boldsymbol{A}_{\overline{ind}}$ and $\boldsymbol{A}_{ind}$ may be observed while $E$ is not available. Regardless, we assume that all attributes, along with the causal features $\boldsymbol{X}_c$, determine the observed features $\boldsymbol{X}$. And the features $\boldsymbol{X}_c$ are the only features that cause $Y$. In the simplest case, we assume no label shift across environments i.e. marginal distribution of $Y$ is constant across train domains and test, $P_{Etr}(y) = P_{Ete}(y)$ (see Figure 2a). More generally, different domains may have different distribution of causal features (in the X-ray example, more women visit one hospital) as shown by $E <-> \boldsymbol{X}_c$ (Figure 2b).

Under the canonical graph, we characterize different kinds of shifts based on the relationship between spurious attributes $\boldsymbol{A}$ and the classification label $Y$. Specifically, $\boldsymbol{A}_{ind}$ is independent of the class label and a change in $P(\boldsymbol{A}_{ind})$ leads to an *Independent* distribution shift. For $\boldsymbol{A}_{\overline{ind}}$, there are

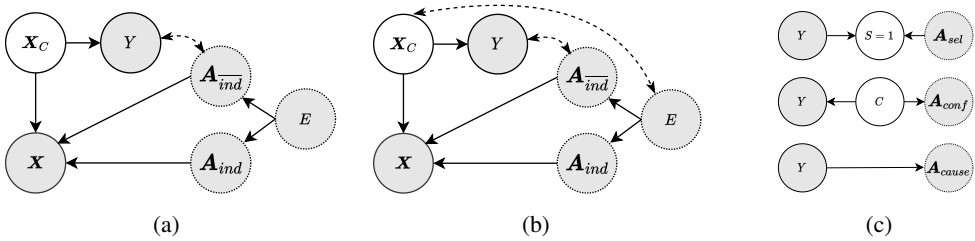

(a)             (b)             (c)

Figure 2: **(a)** Canonical causal graph for specifying *multi-attribute* distribution shifts; **(b)** canonical graph with $E$-$\boldsymbol{X}_c$ correlation. Anti-causal graph shown in Suppl. G. Shaded nodes denote observed variables; since not all attributes may be observed, we use dotted boundary. Dashed lines denote correlation, between $\boldsymbol{X}_c$ and $E$, and $Y$ and $\boldsymbol{A}_{\overline{ind}}$. $E$-$\boldsymbol{X}_c$ correlation can be due to confounding, selection, or causal relationship; all our results hold for any of these relationships (see Suppl. F). **(c)** Different mechanisms for $Y$-$\boldsymbol{A}_{\overline{ind}}$ relationship that lead to *Causal*, *Confounded* and *Selected* shifts.

three mechanisms which can introduce the dashed-line correlation between $A_{\overline{ind}}$ and $Y$ (Figure 2c) – direct-causal ($Y$ causing $A_{\overline{ind}}$), confounding between $Y$ and $A_{\overline{ind}}$ due to a common cause, or selection during the data-generating process. Overall, we define four kinds of shifts based on the causal graph: *Independent*, *Causal*, *Confounded*, and *Selected*[1]. While the canonical graph in Figure 2a is general, resolving each dashed edge into a specific type of shift (causal mechanism) leads to a *realized causal DAG* for a particular dataset. As we shall see, knowledge of these shift types is sufficient to determine the correct independence constraints between observed variables.

Our canonical multi-attribute graph generalizes the DG graph from Mahajan et al. (2021) that considered an *Independent* domain/environment as the only attribute. Under the special case of a single attribute ($|A| = 1$), the canonical graph helps interpret the validity of popular DG methods for a dataset by considering the type of the attribute-label relationship in the data-generating process. For example, let us consider two common constraints in prior work on independence between $\phi$ and a spurious attribute: *unconditional* ($\phi(\boldsymbol{x}) \perp\!\!\!\perp A$) (Veitch et al., 2021; Albuquerque et al., 2020; Ganin et al., 2016) or *conditional* on the label ($\phi(\boldsymbol{x}) \perp\!\!\!\perp A|Y$) (Ghifary et al., 2016; Hu et al., 2019; Li et al., 2018c;d). Under the canonical graph in Figure 2a, the unconditional constraint is true when $A \perp\!\!\!\perp Y$ ($A \in A_{ind}$) but not always for $A_{\overline{ind}}$ (true only under *Confounded* shift). If the relationship is *Causal* or *Selected*, then the conditional constraint is correct. Critically, as Veitch et al. (2021) show for a single-attribute graph, the conditional constraint is not always better; it is an incorrect constraint (not satisfied by $X_c$) under *Confounded* setting. Further, under the canonical graph [with $E$-$X_c$ edge] from Figure 2b, none of these constraints are valid due to a correlation path between $X_c$ and $E$. This shows the importance of considering the generating process for a dataset.

**Inferring attributes-label relationship type.** Whether an attribute belongs to $A_{\overline{ind}}$ or $A_{ind}$ can be learned from data (since $A_{ind} \perp\!\!\!\perp Y$). Under some special conditions with the graph in Figure 2a— assuming all attributes are observed and all attributes in $A_{\overline{ind}}$ are of the same type—we can also identify the type of $A_{\overline{ind}}$ shift: $Y \perp\!\!\!\perp E|A_{\overline{ind}}$ implies *Selected*; if not, then $X \perp\!\!\!\perp E|A_{ind}, A_{\overline{ind}}, Y$ implies *Causal*, otherwise it is *Confounded*. In the general case of Figure 2b, however, it is not possible to differentiate between $A_{cause}$, $A_{conf}$ and $A_{sel}$ using observed data and needs manual input. Fortunately, unlike the full causal graph, the type of relationship between label and an attribute is easier to obtain. For example, in text toxicity classification, toxicity labels are found to be spuriously correlated with certain demographics ($A_{\overline{ind}}$) (Dixon et al., 2018; Koh et al., 2021; Park et al., 2018); while in medical applications where data is collected from small number of hospitals, shifts arise due to different methods of slide staining and image acquisition ($A_{ind}$) (Koh et al., 2021; Komura & Ishikawa, 2018; Tellez et al., 2019). Suppl. A contains additional real-world examples with attributes.

## 3.2 Independence constraints depend on attribute↔label relationship

We list the independence constraints between $\langle X_c, A, Y \rangle$ under the canonical graphs from Figure 2, which can be used to derive the correct regularization constraints to be applied on $\phi$ (Theorem 2.1).

**Proposition 3.1.** *Given a causal DAG realized by specifying the target-attributes relationship in Figure 2a, the correct constraint depends on the relationship of label $Y$ with the attributes $A$. As shown, $A$ can be split into $A_{\overline{ind}}$, $A_{ind}$ and $E$, where $A_{\overline{ind}}$ can be further split into subsets that have a causal ($A_{cause}$), confounded ($A_{conf}$), selected ($A_{sel}$) relationship with $Y$ ($A_{\overline{ind}} = A_{cause} \cup A_{conf} \cup A_{sel}$). Then, the (conditional) independence constraints $X_c$ should satisfy are,*

1. *Independent:* $X_c \perp\!\!\!\perp A_{ind}$; $X_c \perp\!\!\!\perp E$; $X_c \perp\!\!\!\perp A_{ind}|Y$; $X_c \perp\!\!\!\perp A_{ind}|E$; $X_c \perp\!\!\!\perp A_{ind}|Y, E$
2. *Causal:* $X_c \perp\!\!\!\perp A_{cause}|Y$; $X_c \perp\!\!\!\perp E$; $X_c \perp\!\!\!\perp A_{cause}|Y, E$
3. *Confounded:* $X_c \perp\!\!\!\perp A_{conf}$; $X_c \perp\!\!\!\perp E$; $X_c \perp\!\!\!\perp A_{conf}|E$
4. *Selected:* $X_c \perp\!\!\!\perp A_{sel}|Y$; $X_c \perp\!\!\!\perp A_{sel}|Y, E$

**Corollary 3.1.** *All the above derived constraints are valid for Graph 2a. However, in the presence of a correlation between $E$ and $X_c$ (Graph 2b), only the constraints conditioned on $E$ hold true.*

Corollary 3.1 implies that if we are not sure about $E$-$X_c$ correlation, $E$-*conditioned* constraints should be used. By considering independence constraints over *attributes* that may represent any observed variable, our graph-based characterization unites the single-domain (group-wise) (Sagawa et al., 2020) and multi-domain generalization tasks. Whether attributes represent auxiliary attributes, group indicators, or data sources, Proposition 3.1 provides the correct regularization constraint.

---

[1]Note that for *Selected* to satisfy the assumptions of Theorem 2.1 implies that $X_c$ is fully predictive of $Y$ or that the noise in $Y$-$X_c$ relationship is independent of the features driving the selection process.

### 3.3 A FIXED CONDITIONAL INDEPENDENCE CONSTRAINT CANNOT WORK FOR ALL SHIFTS

Combining with Theorem 2.1, Proposition 3.1 shows that the necessary constraints for a risk-invariant predictor's representation $\phi(\boldsymbol{X})$ are different for different types of attributes. This leads us to our key result: under multi-attribute shifts, a single (conditional) independence constraint cannot be valid for all kinds of shifts. Remarkably, this result is true even for single-attribute shifts: any algorithm with a fixed conditional independence constraint (e.g., as listed in Table 1 (Gretton et al., 2012; Arjovsky et al., 2019; Li et al., 2018b; Sun & Saenko, 2016)) cannot work for all datasets.

**Theorem 3.1.** *Under the canonical causal graph in Figure 2(a,b), there exists no (conditional) independence constraint over $\langle \boldsymbol{X}_c, \boldsymbol{A}, Y \rangle$ that is valid for all realized DAGs as the type of multi-attribute shifts vary. Hence, for any predictor algorithm for $Y$ that uses a single (conditional) independence constraint over its representation $\phi(\boldsymbol{X})$, $\boldsymbol{A}$ and $Y$, there exists a realized DAG $\mathcal{G}$ and a corresponding training dataset such that the learned predictor cannot be a risk-invariant predictor for distributions in $\mathcal{P}_\mathcal{G}$, where $\mathcal{P}_\mathcal{G}$ is the set of distributions obtained by changing $P(\boldsymbol{A}|Y)$.*

**Corollary 3.2.** *Even when $|\boldsymbol{A}| = 1$, an algorithm using a single independence constraint over $\langle \phi(\boldsymbol{X}), A, Y \rangle$ cannot yield a risk-invariant predictor for all kinds of single-attribute shift datasets.*

Corollary 3.2 adds theoretical evidence for past empirical demonstrations of inconsistent performance of DG algorithms (Wiles et al., 2022; Ye et al., 2022).To demonstrate its significance, we provide OoD generalization results on a simple "slab" setup (Shah et al., 2020) with three datasets (*Causal*, *Confounded*, and *Selected* shifts) in Suppl. E.2. We evaluate two constraints motivated by DG literature Mahajan et al. (2021): unconditional $\boldsymbol{X}_c \perp\!\!\!\perp \boldsymbol{A}|E$, and conditional on label $\boldsymbol{X}_c \perp\!\!\!\perp \boldsymbol{A}|Y, E$. As predicted by Corollary 3.2, neither constraint obtains best accuracy on all three datasets (Table 6).

## 4 CAUSALLY ADAPTIVE CONSTRAINT MINIMIZATION (*CACM*)

Motivated by Sec. 3, we present *CACM*, an algorithm that adaptively chooses regularizing constraints for multi-attribute shift datasets (full algorithm for any general DAG in Suppl. C). It has two phases.

**Phase I. Derive correct independence constraints.** If a dataset's DGP satisfies the canonical graph, *CACM* requires a user to specify the relationship type for each attribute and uses the constraints from Proposition 3.1. For other datasets, *CACM* requires a causal graph describing the dataset's DGP and uses the following steps to derive the independence constraints. Let $\mathcal{V}$ be the set of observed variables in the graph except $Y$, and $\mathcal{C}$ be the list of constraints.

1. For each observed variable $V \in \mathcal{V}$, check whether $(\boldsymbol{X}_c, V)$ are $d$-separated. Add $\boldsymbol{X}_c \perp\!\!\!\perp V$ to $\mathcal{C}$.
2. If not, check if $(\boldsymbol{X}_c, V)$ are $d$-separated conditioned on any subset $Z$ of the remaining observed variables in $\mathcal{Z} = \{Y\} \cup \mathcal{V} \setminus \{V\}$. For each subset $Z$ with $d$-separation, add $\boldsymbol{X}_c \perp\!\!\!\perp V|Z$ to $\mathcal{C}$.

**Phase II. Apply regularization penalty using derived constraints.** In Phase II, *CACM* applies those constraints as a regularizer to the standard ERM loss, $g_1, \phi = \arg\min_{g_1,\phi};\quad \ell(g_1(\phi(\boldsymbol{x})), y) + RegPenalty$, where $\ell$ is cross-entropy loss. The regularizer optimizes for valid constraints over all observed variables $V \in \mathcal{V}$. Below we provide the regularizer term for datasets following the canonical graphs from Figure 2 ($\mathcal{V} = \boldsymbol{A}$). We choose Maximum Mean Discrepancy (MMD) (Gretton et al., 2012) to apply our penalty (in principle, any metric for conditional independence would work).

Since $\boldsymbol{A}$ includes multiple attributes, the regularizer penalty depends on the type of distribution shift for each attribute. For instance, for $A \in \boldsymbol{A}_{ind}$ (*Independent*), to enforce $\phi(\boldsymbol{x}) \perp\!\!\!\perp A$, we aim to minimize the distributional discrepancy between $P(\phi(\boldsymbol{x})|A = a_i)$ and $P(\phi(\boldsymbol{x})|A = a_j)$, for all $i, j$ values of $A$. However, since the same constraint is applicable on $E$, it is statistically efficient to apply the constraint on $E$ (if available) as there may be multiple closely related values of $A$ in a domain (e.g., slide stains collected from one hospital may be spread over similar colors, but not exactly the same). Hence, we apply the constraint on distributions $P(\phi(\boldsymbol{x})|E = E_i)$ and $P(\phi(\boldsymbol{x})|E = E_j)$ if $E$ is observed (and $A$ may/may not be unobserved), otherwise we apply the constraint over $A$.

$$RegPenalty_{\boldsymbol{A}_{ind}} = \sum_{i=1}^{|\boldsymbol{A}_{ind}|} \sum_{j>i} \mathrm{MMD}(P(\phi(\boldsymbol{x})|a_{i,ind}), P(\phi(\boldsymbol{x})|a_{j,ind})) \tag{2}$$

For $A \in \boldsymbol{A}_{cause}$ (*Causal*), following Proposition 3.1, we consider distributions $P(\phi(\boldsymbol{x})|A = a_i, Y = y)$ and $P(\phi(\boldsymbol{x})|A = a_j, Y = y)$. We additionally condition on $E$ as there may be a correlation

between $E$ and $\boldsymbol{X}_c$ (Figure 2b), which renders other constraints incorrect (Corollary 3.1). We similarly obtain regularization terms for *Confounded* and *Selected* (Suppl. C).

$$RegPenalty_{\boldsymbol{A}_{cause}} = \sum_{|E|} \sum_{y \in Y} \sum_{i=1}^{|\boldsymbol{A}_{cause}|} \sum_{j>i} \mathrm{MMD}(P(\phi(\boldsymbol{x})|a_{i,cause},y), P(\phi(\boldsymbol{x})|a_{j,cause},y))$$

The final $RegPenalty$ is a sum of penalties over all attributes, $RegPenalty = \sum_{A \in \boldsymbol{A}} \lambda_A Penalty_A$, where $\lambda_A$ are the hyperparameters. Unlike prior work (Makar et al., 2022; Veitch et al., 2021), we do not restrict ourselves to binary-valued attributes and classes.

***CACM*'s relationship with existing DG algorithms.** Table 1 shows common constraints used by popular DG algorithms. *CACM*'s strength lies in *adaptively* selecting constraints based on the causal relationships in the DGP. Thus, depending on the dataset, applying *CACM* for a single-attribute shift may involve applying the same constraint as in MMD or C-MMD algorithms. For example, in Rotated-MNIST dataset with $E = A_{ind} = rotation$, the effective constraint for MMD, DANN, CORAL algorithms ($\phi \perp\!\!\!\perp E$) is the same as *CACM*'s constraint $\phi \perp\!\!\!\perp A_{ind}$ for *Independent* shift.

## 5 EMPIRICAL EVALUATION

We perform experiments on MNIST, small NORB, and Waterbirds datasets to demonstrate our main claims: existing DG algorithms perform worse on multi-attribute shifts; *CACM* with the correct graph-based constraints significantly outperforms these algorithms; and incorrect constraints cannot match the above accuracy. While we provide constraints for all shifts in Proposition 3.1, our empirical experiments with datasets focus on commonly occurring *Causal* and *Independent* shifts. All experiments are performed in PyTorch 1.10 with NVIDIA Tesla P40 and P100 GPUs, and building on DomainBed (Gulrajani & Lopez-Paz, 2021) and OoD-Bench (Ye et al., 2022). Regularizing on model's logit scores provides better accuracy than $\phi(\boldsymbol{x})$; hence we adopt it for all our experiments.

### 5.1 DATASETS & BASELINE DG ALGORITHMS

We introduce three new datasets for the multi-attribute shift problem. For all datasets, details of environments, architectures, visualizations, and setup generation are in Suppl. D.1.

**MNIST.** Colored (Arjovsky et al., 2019) and Rotated MNIST (Ghifary et al., 2015) present *Causal* ($\boldsymbol{A}_{cause} = color$) and *Independent* ($\boldsymbol{A}_{ind} = rotation$) distribution shifts, respectively. We combine these to obtain a multi-attribute dataset with $\boldsymbol{A}_{cause}$ and $\boldsymbol{A}_{ind}$ ($col + rot$). For comparison, we also evaluate on single-attribute $\boldsymbol{A}_{cause}$ (Colored) and $\boldsymbol{A}_{ind}$ (Rotated) MNIST datasets.

**small NORB** (LeCun et al., 2004). This dataset was used by Wiles et al. (2022) to create a challenging DG task with single-attribute shifts, having multi-valued classes and attributes over realistic 3D objects. We create a multi-attribute shift dataset ($light + azi$), consisting of a causal connection, $\boldsymbol{A}_{cause} = lighting$, between lighting and object category $Y$; and $\boldsymbol{A}_{ind} = azimuth$ that varies independently across domains. We also evaluate on single-attribute $\boldsymbol{A}_{cause}$ (*lighting*) and $\boldsymbol{A}_{ind}$ (*azimuth*) datasets.

**Waterbirds.** We use the original dataset (Sagawa et al., 2020) where bird type (water or land) ($Y$) is spuriously correlated with background ($\boldsymbol{A}_{cause}$). To create a multi-attribute setup, we add different weather effects ($\boldsymbol{A}_{ind}$) to train and test data with probability $p = 0.5$ and $1.0$ respectively.

**Baseline DG algorithms & implementation.** We consider baseline algorithms optimizing for different constraints and statistics to compare to causal adaptive regularization: IRM (Arjovsky et al., 2019), IB-ERM and IB-IRM (Ahuja et al., 2021), VREx (Krueger et al., 2021), MMD (Li et al., 2018b), CORAL (Sun & Saenko, 2016), DANN (Gretton et al., 2012), Conditional-MMD (C-MMD) (Li et al., 2018b), Conditional-DANN (CDANN) (Li et al., 2018d), GroupDRO (Sagawa et al., 2020), Mixup (Yan et al., 2020), MLDG (Li et al., 2018a), SagNet (Nam et al., 2021), and RSC (Huang et al., 2020). Following DomainBed (Gulrajani & Lopez-Paz, 2021), a random search is performed 20 times over the hyperparameter distribution for 3 seeds. The best models obtained across the three seeds are used to compute the mean and standard error. We use a validation set that follows the test domain distribution consistent with previous work on these datasets (Arjovsky et al., 2019; Sagawa et al., 2020; Wiles et al., 2022; Ye et al., 2022). Further details are in Suppl. D.

Table 2: **Colored + Rotated MNIST:** Accuracy on unseen domain for singe-attribute ($color, rotation$) and multi-attribute ($col + rot$) distribution shifts; **small NORB:** Accuracy on unseen domain for single-attribute ($lighting, azimuth$) and multi-attribute ($light + azi$) distribution shifts. **Waterbirds:** Worst-group accuracy on unseen domain for single- and multi-attribute shifts.

| Algo | Colored+Rotated MNIST Accuracy | | | small NORB Accuracy | | | Waterbirds Worst-group accuracy | |
|---|---|---|---|---|---|---|---|---|
| | *color* | *rotation* | *col+rot* | *lighting* | *azimuth* | *light+azi* | original | multi-attr |
| ERM | 30.9 ±1.6 | 61.9 ±0.5 | 25.2 ±1.3 | 65.5 ±0.7 | 78.6 ±0.7 | 64.0 ±1.2 | $66.0 \pm 3.7$ | $37.0 \pm 1.1$ |
| IB-ERM | 27.8 ±0.7 | 62.1 ±0.8 | 41.2 ±4.1 | 66.0 ±0.9 | 75.9 ±1.2 | 61.2 ±0.1 | $66.9 \pm 4.6$ | $40.8 \pm 5.6$ |
| IRM | 50.0 ±0.1 | 61.2 ±0.3 | 39.6 ±6.7 | 66.7 ±1.5 | 75.7 ±0.4 | 61.7 ±0.5 | $61.2 \pm 5.2$ | $37.7 \pm 1.7$ |
| IB-IRM | 49.9 ±0.1 | 61.4 ±0.9 | 49.3 ±0.3 | 64.7 ±0.8 | 77.6 ±0.3 | 62.2 ±1.2 | $62.3 \pm 7.7$ | $46.9 \pm 6.5$ |
| VREx | 30.3 ±1.6 | 62.1 ±0.4 | 23.3 ±0.4 | 64.7 ±1.0 | 77.6 ±0.5 | 62.5 ±1.6 | $68.8 \pm 2.5$ | $38.1 \pm 2.3$ |
| MMD | 29.7 ±1.8 | 62.2 ±0.5 | 24.1 ±0.6 | 66.6 ±1.6 | 76.7 ±1.1 | 62.5 ±0.3 | $68.1 \pm 4.4$ | $45.2 \pm 2.4$ |
| CORAL | 28.5 ±0.8 | **62.5 ±0.7** | 23.5 ±1.1 | 64.7 ±0.5 | 77.2 ±0.7 | 62.9 ±0.3 | $73.6 \pm 4.8$ | $54.1 \pm 3.0$ |
| DANN | 20.7 ±0.8 | 61.9 ±0.7 | 32.0 ±7.8 | 64.6 ±1.4 | 78.6 ±0.7 | 60.8 ±0.7 | $78.5 \pm 1.8$ | $55.5 \pm 4.6$ |
| C-MMD | 29.4 ±0.2 | 62.3 ±0.4 | 32.2 ±7.0 | 65.8 ±0.8 | 76.9 ±1.0 | 61.0 ±0.9 | $77.0 \pm 1.2$ | $52.3 \pm 1.9$ |
| CDANN | 30.8 ±8.0 | 61.8 ±0.2 | 32.2 ±7.0 | 64.9 ±0.5 | 77.3 ±0.3 | 60.8 ±0.9 | $69.9 \pm 3.3$ | $49.7 \pm 3.9$ |
| DRO | 33.9 ±0.4 | 60.6 ±0.9 | 25.3 ±0.5 | 65.5 ±0.7 | 77.1 ±1.0 | 62.3 ±0.6 | $70.4 \pm 1.3$ | $53.1 \pm 2.2$ |
| Mixup | 25.1 ±1.2 | 61.4 ±0.6 | 21.1 ±1.6 | 66.2 ±1.3 | 80.4 ±0.5 | 57.1 ±1.5 | $74.2 \pm 3.9$ | $64.7 \pm 2.4$ |
| MLDG | 31.0 ±0.3 | 61.6 ±0.8 | 24.4 ±0.7 | 66.0 ±0.7 | 77.9 ±0.5 | 64.2 ±0.6 | $70.8 \pm 1.5$ | $34.5 \pm 1.7$ |
| SagNet | 28.2 ±0.8 | 60.7 ±0.7 | 23.7 ±0.2 | 65.9 ±1.5 | 76.1 ±0.4 | 62.2 ±0.5 | $69.1 \pm 1.0$ | $40.6 \pm 7.1$ |
| RSC | 29.1 ±1.9 | 62.3 ±0.4 | 22.8 ±0.3 | 62.4 ±0.4 | 75.6 ±0.6 | 61.8 ±1.3 | $64.6 \pm 6.5$ | $40.9 \pm 3.6$ |
| *CACM* | **70.4 ±0.5** | 62.4 ±0.4 | **54.1 ±1.3** | **85.4 ±0.5** | **80.5 ±0.6** | **69.6 ±1.6** | $\mathbf{84.5 \pm 0.6}$ | $\mathbf{70.5 \pm 1.1}$ |

Table 3: small NORB *Causal* shift. Comparing $\boldsymbol{X}_c \perp\!\!\!\perp \boldsymbol{A}_{cause} |Y, E$ with incorrect constraints.

| Constraint | Accuracy |
|---|---|
| $\boldsymbol{X}_c \perp\!\!\!\perp \boldsymbol{A}_{cause}$ | $72.7 \pm 1.1$ |
| $\boldsymbol{X}_c \perp\!\!\!\perp \boldsymbol{A}_{cause} |E$ | $76.2 \pm 0.9$ |
| $\boldsymbol{X}_c \perp\!\!\!\perp \boldsymbol{A}_{cause} |Y$ | $79.7 \pm 0.9$ |
| $\boldsymbol{X}_c \perp\!\!\!\perp \boldsymbol{A}_{cause} |Y, E$ | $\mathbf{85.4 \pm 0.5}$ |

Table 4: Comparing $\boldsymbol{X}_c \perp\!\!\!\perp \boldsymbol{A}_{cause} |Y, E$ and $\boldsymbol{X}_c \perp\!\!\!\perp \boldsymbol{A}_{cause} |Y$ for *Causal* shift in MNIST and small NORB. The constraint implied by $E$-$\boldsymbol{X}_c$ correlation (Fig. 2b, Prop. 3.1) affects accuracy.

| Constraint | MNIST | small NORB |
|---|---|---|
| $\boldsymbol{X}_c \perp\!\!\!\perp \boldsymbol{A}_{cause} |Y$ | $69.7 \pm 0.2$ | $79.7 \pm 0.9$ |
| $\boldsymbol{X}_c \perp\!\!\!\perp \boldsymbol{A}_{cause} |Y, E$ | $70.4 \pm 0.5$ | $85.4 \pm 0.5$ |

## 5.2 RESULTS

**Correct constraint derived from the causal graph matters.** Table 2 shows the accuracy on test domain for all datasets. Comparing the three prediction tasks for MNIST and small NORB, for all algorithms, accuracy on unseen test domain is highest under $\boldsymbol{A}_{ind}$ shift and lowest under two-attribute shift ($\boldsymbol{A}_{ind} \cup \boldsymbol{A}_{cause}$), reflecting the difficulty of a multi-attribute distribution shift. On the two-attribute shift task in MNIST, all DG algorithms obtain less than 50% accuracy whereas *CACM* obtains a 5% absolute improvement. Results on small NORB dataset are similar: *CACM* obtains 69.6% accuracy on the two-attribute task while the nearest baseline is MLDG at 64.2%.

On both MNIST and small NORB, *CACM* also obtains highest accuracy on the $\boldsymbol{A}_{cause}$ task. On MNIST, even though IRM and VREx have been originally evaluated for the Color-only ($\boldsymbol{A}_{cause}$) task, under an extensive hyperparameter sweep as recommended in past work (Gulrajani & Lopez-Paz, 2021; Krueger et al., 2021; Ye et al., 2022), we find that *CACM* achieves a substantially higher accuracy (70%) than these methods, just 5 units lower than the optimal 75%. While the $\boldsymbol{A}_{ind}$ task is relatively easier, algorithms optimizing for the correct constraint achieve highest accuracy. Note that MMD, CORAL, DANN, and *CACM* are based on the same independence constraint (see Table 1). As mentioned in Section 4, we use the the domain attribute $E$ for *CACM*'s regularization constraint for $\boldsymbol{A}_{ind}$ task, for full comparability with other algorithms that also use $E$. The results indicate the importance of adaptive regularization for generalization.

Table 2 also shows the OoD accuracy of algorithms on the original Waterbirds dataset (Sagawa et al., 2020) and its multi-attribute shift variant. Here we evaluate using worst-group accuracy consistent with past work (Sagawa et al., 2020; Yao et al., 2022). We observe that on single-attribute ($\boldsymbol{A}_{cause}$) as well as multi-attribute shift, *CACM* significantly outperforms baselines ($\sim$6% absolute improvement).

**Incorrect constraints hurt generalization.** We now directly compare the effect of using correct versus *incorrect* (but commonly used) constraints for a dataset. To isolate the effect of a single constraint, we first consider the single-attribute shift on $A_{cause}$ and compare the application of different regularizer constraints. Proposition 3.1 provides the correct constraint for $A_{cause}$: $X_c \perp\!\!\!\perp A_{cause} | Y, E$. In addition, using $d$-separation on *Causal*-realized DAG from Figure 2, we see the following invalid constraints, $X_c \perp\!\!\!\perp A_{cause} | E$, $X_c \perp\!\!\!\perp A_{cause}$. Without knowing that the DGP corresponds to a *Causal* shift, one may apply these constraints that do not condition on the class. Results on small NORB (Table 3) show that using the incorrect constraint has an adverse effect: the correct constraint yields 85% accuracy while the best incorrect constraint achieves 79.7%. Moreover, unlike the correct constraint, application of the incorrect constraint is sensitive to the $\lambda$ (regularization weight) parameter: as $\lambda$ increases, accuracy drops to less than 40% (Suppl. E.3, Figure 7).

Comparing these constraints on small NORB and MNIST (Table 4) reveals the importance of making the right structural assumptions. Typically, DG algorithms assume that distribution of causal features $X_c$ does not change across domains (as in the graph in Fig. 2a). Then, both $X_c \perp\!\!\!\perp A_{cause} | Y, E$ and $X_c \perp\!\!\!\perp A_{cause} | Y$ should be correct constraints. However, conditioning on both $Y$ and $E$ provides a 5% point gain over conditioning on $Y$ in NORB while the accuracy is comparable for MNIST. Information about the data-generating process explains the result: Different domains in MNIST include samples from the same distribution whereas small NORB domains are sampled from a different set of toy objects, thus creating a correlation between $X_c$ and $E$, corresponding to the graph in Fig. 2b. Without information on the correct DGP, such gains will be difficult.

Finally, we replicate the above experiment for the multi-attribute shift setting for small NORB. To construct an incorrect constraint, we interchange the variables before inputting to *CACM* algorithm ($A_{ind}$ gets used as $A_{cause}$ and vice-versa). Accuracy with interchanged variables ($65.1 \pm 1.6$) is lower than that of correct *CACM* ($69.6 \pm 1.6$). More ablations where baseline DG algorithms are provided *CACM*-like attributes as environment are in Suppl. E.4.

## 6 Related Work

Improving the robustness of models in the face of distribution shifts is a key challenge. Several works have attempted to tackle the domain generalization problem (Wang et al., 2021; Zhou et al., 2021) using different approaches – data augmentation (Cubuk et al., 2020; He et al., 2016; Zhu et al., 2017), and representation learning (Arjovsky et al., 2019; Deng et al., 2009; Higgins et al., 2017) being popular ones. Trying to gauge the progress made by these approaches, Gulrajani and Lopez-Paz (Gulrajani & Lopez-Paz, 2021) find that existing state-of-the-art DG algorithms do not improve over ERM. More recent work (Wiles et al., 2022; Ye et al., 2022) uses datasets with different single-attribute shifts and empirically shows that different algorithms perform well over different distribution shifts, but no single algorithm performs consistently across all. We provide (1) multi-attribute shift benchmark datasets; (2) a causal interpretation of different kinds of shifts; and (3) an adaptive algorithm to identify the correct regularizer. While we focus on images, OoD generalization on graph data is also challenged by multiple types of distribution shifts (Chen et al., 2022).

**Causally-motivated learning.** There has been recent work focused on *causal representation learning* (Arjovsky et al., 2019; Krueger et al., 2021; Locatello et al., 2020; Schölkopf et al., 2021) for OoD generalization. While these works attempt to learn the constraints for causal features from input features, we show that it is necessary to model the data-generating process and have access to auxiliary attributes to obtain a risk-invariant predictor, especially in *multi-attribute* distribution shift setups. Recent research has shown how causal graphs can be used to characterize and analyze the different kinds of distribution shifts that occur in real-world settings (Makar et al., 2022; Veitch et al., 2021). Our approach is similar in motivation but we extend from single-domain, single-attribute setups in past work to formally introduce *multi-attribute* distribution shifts in more complex and real-world settings. Additionally, we do not restrict ourselves to binary-valued classes and attributes.

## 7 Discussion

We introduced *CACM*, an adaptive OoD generalization algorithm to characterize *multi-attribute* shifts and apply the correct independence constraints. Through empirical experiments and theoretical analysis, we show the importance of modeling the causal relationships in the data-generating process. That said, our work has limitations: the constraints from *CACM* are necessary but not sufficient for a risk-invariant predictor (e.g., unable to remove influence of unobserved spurious attributes).

## 8    ETHICS AND BROADER IMPACT STATEMENT

Our work on modeling the data-generating process for improved out-of-distribution generalization is an important advance in building robust predictors for practical settings. Such prediction algorithms, including methods building on representation learning, are increasingly a key element of decision-support and decision-making systems. We expect our approach to creating a robust predictor to be particularly valuable in real world setups where *spurious* attributes and real-world multi-attribute settings lead to biases in data. While not the focus of this paper, *CACM* may be applied to mitigate social biases (e.g., in language and vision datasets) whose structures can be approximated by the graphs in Figure 2. Risks of using methods such as *CACM*, include excessive reliance or a false sense of confidence. While methods such as *CACM* ease the process of building robust models, there remain many ways that an application may still fail (e.g., incorrect structural assumptions). AI applications must still be designed appropriately with support of all stakeholders and potentially affected parties, tested in a variety of settings, etc.

## 9    REPRODUCIBILITY STATEMENT

We provide all required experimental details in Suppl. D including dataset details, training details and hyperparameter search sweeps. We additionally submit our code as part of supplementary material which can be used to reproduce the experiments. We provide a demo notebook for prediction using *CACM* in the DoWhy[2] library. We provide proofs for all our theoretical results in Suppl. B.

## 10    ACKNOWLEDGEMENTS

We thank Abhinav Kumar, Adith Swaminathan, Yiding Jiang, and Dhruv Agarwal for helpful feedback and comments on the draft. We would also like to thank the anonymous reviewers for their valuable feedback.

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

## A   PRESENCE OF AUXILIARY ATTRIBUTE INFORMATION IN DATASETS

Unlike the full causal graph, attribute values as well as the relationships between class labels and attributes is often known. *CACM* assumes access to attribute labels $\boldsymbol{A}$ only during training time, which are collected as part of the data collection process (e.g., as metadata with training data (Makar et al., 2022)). We start by discussing the availability of attributes in WILDS (Koh et al., 2021), a set of real-world datasets adapted for the domain generalization setting. Attribute labels available in the datasets include, the *time (year)* and *region* associated with satellite images in FMoW dataset (Christie et al., 2018) for predicting land use category, *hospital* from where the tissue patch was collected for tumor detection in Camelyon17 dataset (Bandi et al., 2018) and the *demographic* information for CivilComments dataset (Borkan et al., 2019). (Koh et al., 2021) create different domains in WILDS using this metadata, consistent with our definition of $E \in \boldsymbol{A}$ as a special domain attribute.

In addition, *CACM* requires the type of relationship between label $Y$ and attributes. This is often known, either based on how the dataset was collected or inferred based on domain knowledge or observation. While the distinction between $\boldsymbol{A}_{ind}$ and $\boldsymbol{A}_{\overline{ind}}$ can be established using a statistical test of independence on a given dataset, in general, the distinction between $\boldsymbol{A}_{cause}$, $\boldsymbol{A}_{sel}$ and $\boldsymbol{A}_{conf}$ within $\boldsymbol{A}_{\overline{ind}}$ needs to be provided by the user. As we show for the above datasets, the type of relationship can be inferred based on common knowledge or information on how the dataset was collected.

For FMoW dataset, *time* can be considered an *Independent* attribute ($\boldsymbol{A}_{ind}$) since it reflects the time at which images are captured which is not correlated with $Y$; whereas *region* is a *Confounded* attribute since certain regions associated with certain $Y$ labels are over-represented due to ease of data collection. Note that region cannot lead to *Causal* shift since the decision to take images in a region was not determined by the final label nor *Selected* for the same reason that the decision was not taken based on values of $Y$. Similarly, for the Camelyon17 dataset, it is known that differences in slide staining or image acquisition leads to variation in tissue slides across *hospitals*, thus implying that *hospital* is an *Independent* attribute ($\boldsymbol{A}_{ind}$) (Koh et al., 2021; Komura & Ishikawa, 2018; Tellez et al., 2019); As another example from healthcare, a study in MIT Technology Review[3] discusses biased data where a person's *position* ($\boldsymbol{A}_{conf}$) was spuriously correlated with disease prediction as patients lying down were more likely to be ill. As another example, (Sagawa et al., 2020) adapt MultiNLI dataset for OoD generalization due to the presence of spurious correlation between *negation words* (attribute) and the contradiction label between "premise" and "hypothesis" inputs. Here, negation words are a result of the contradiction label (*Causal* shift), however this relationship between negation words and label may not always hold. Finally, for the CivilComments dataset, we expect the *demographic* features to be *Confounded* attributes as there could be biases which result in spurious correlation between comment toxicity and demographic information.

To provide examples showing the availability of attributes and their type of relationship with the label, Table 5 lists some popular datasets used for DG and the associated auxiliary information present as metadata. In addition to above discussed datasets, weinclude the popularly used Waterbirds dataset (Sagawa et al., 2020) where the type of *background* (land/water) is assigned to bird images based on bird label; hence, being a *Causal* attribute (results on Waterbirds dataset are in Table 2).

Table 5: Commonly used DG datasets include auxiliary information.

| Dataset | Attribute(s) | $Y - A$ relationship |
| --- | :---: | :---: |
| FMoW-WILDS (Koh et al., 2021) | time | $\boldsymbol{A}_{ind}$ |
| | region | $\boldsymbol{A}_{conf}$ |
| Camelyon17-WILDS (Koh et al., 2021) | hospital | $\boldsymbol{A}_{ind}$ |
| Waterbirds (Sagawa et al., 2020) | background (land/water) | $\boldsymbol{A}_{cause}$ |
| MultiNLI (Sagawa et al., 2020) | negation word | $\boldsymbol{A}_{cause}$ |
| CivilComments-WILDS (Koh et al., 2021) | demographic | $\boldsymbol{A}_{conf}$ |

---

[3]`https://www.technologyreview.com/2021/07/30/1030329/machine-learning-ai-failed-covid-hospital-diagnosis-pandemic/`

# B  Proofs

## B.1  Proof of Theorem 2.1

**Theorem 2.1.** *Consider a causal DAG $\mathcal{G}$ over $\langle \boldsymbol{X}_c, \boldsymbol{X}, \boldsymbol{A}, Y \rangle$ and a corresponding generated dataset $(\boldsymbol{x}_i, \boldsymbol{a}_i, y_i)_{i=1}^{n}$, where $\boldsymbol{X}_c$ is unobserved. Assume that graph $\mathcal{G}$ has the following property: $\boldsymbol{X}_c$ is defined as the set of all parents of $Y$ ($\boldsymbol{X}_c \to Y$); and $\boldsymbol{X}_c, \boldsymbol{A}$ together cause $\boldsymbol{X}$ ($\boldsymbol{X}_c \to \boldsymbol{X}$, and $\boldsymbol{A} \to \boldsymbol{X}$). The graph may have any other edges (see, e.g., DAG in Figure 1(b)). Let $\mathcal{P}_\mathcal{G}$ be the set of distributions consistent with graph $\mathcal{G}$, obtained by changing $P(\boldsymbol{A}|Y)$ but not $P(Y|\boldsymbol{X}_c)$. Then the conditional independence constraints satisfied by $\boldsymbol{X}_c$ are necessary for a (cross-entropy) risk-invariant predictor over $\mathcal{P}_\mathcal{G}$. That is, if a predictor for $Y$ does not satisfy any of these constraints, then there exists a data distribution $P' \in \mathcal{P}_\mathcal{G}$ such that predictor's risk will be higher than its risk in other distributions.*

*Proof.* We consider $\boldsymbol{X}, Y, \boldsymbol{X}_c, \boldsymbol{A}$ as random variables that are generated according to the data-generating process corresponding to causal graph $\mathcal{G}$. We assume that $\boldsymbol{X}_c$ represents all the parents of $Y$. $\boldsymbol{X}_c$ also causes the observed features $\boldsymbol{X}$ but $\boldsymbol{X}$ may be additionally affected by the attributes $\boldsymbol{A}$. Let $\hat{y} = g(\boldsymbol{x})$ be a candidate predictor. Then $g(\boldsymbol{X})$ represents a random vector based on a deterministic function $g$ of $\boldsymbol{X}$.

Suppose there is an independence constraint $\psi$ that is satisfied by $\boldsymbol{X}_c$ but not $g(\boldsymbol{X})$. [4] Below we show that such a predictor $g$ is not risk-invariant: there exist two data distributions with different $P(\boldsymbol{A}|Y)$ such that the risk of $g$ is different for them.

Without loss of generality, we can write $g(\boldsymbol{x})$ as,

$$g(\boldsymbol{x}) = (g(\boldsymbol{x})/h(\boldsymbol{x}_c)) * h(\boldsymbol{x}_c) = g'(\boldsymbol{x}, \boldsymbol{x}_c) h(\boldsymbol{x}_c) \qquad \forall \boldsymbol{x} \tag{3}$$

where $h$ is an arbitrary, non-zero, deterministic function of the random variable $\boldsymbol{X}_c$. Since $\boldsymbol{X}_c$ satisfies the (conditional) independence constraint $\psi$ and $h$ is a deterministic function, $h(\boldsymbol{X}_c)$ also satisfies $\psi$. Also since the predictor $g(\boldsymbol{X})$ does not satisfy the constraint $\psi$, it implies that the random vector $g'(\boldsymbol{X}, \boldsymbol{X}_c)$ cannot satisfy the constraint $\psi$. Thus, $g'(\boldsymbol{X}, \boldsymbol{X}_c)$ cannot be a function of $\boldsymbol{X}_c$ only; it needs to depend on $\boldsymbol{X}$ too. Since $\boldsymbol{X}$ has two parents in the causal graph, $\boldsymbol{X}_c$ and $\boldsymbol{A}$, this implies that $g'(\boldsymbol{X}, \boldsymbol{X}_c)$ must depend on $\boldsymbol{A}$ too, and hence $g'(\boldsymbol{X}, \boldsymbol{X}_c)$ and $\boldsymbol{A}$ are not independent.

Now, let us construct two data distributions $P_1$ and $P_2$ with the same marginal distributions of $P(Y)$, $P(\boldsymbol{A})$ and $P(\boldsymbol{X}_c)$, such that $P(\boldsymbol{A}|Y)$ changes across them. Note that $P(Y|\boldsymbol{X}_c)$ stays invariant because of the independent and stable causal mechanism property, i.e., $P_1(Y|\boldsymbol{X}_c) = P_2(Y|\boldsymbol{X}_c)$. For these two data distributions, change in $P(\boldsymbol{A}|Y)$ implies a change in $P(Y|\boldsymbol{A})$, i.e., $P_1(Y|\boldsymbol{A}) \neq P_2(Y|\boldsymbol{A})$, since $P(Y|\boldsymbol{A}) = P(\boldsymbol{A}|Y)P(Y)/P(\boldsymbol{A})$. Also, since $g'(\boldsymbol{X}, \boldsymbol{X}_c)$ and $\boldsymbol{A}$ are not independent, $P(Y|g'(\boldsymbol{X}, \boldsymbol{X}_c))$ will change, i.e., $P_1(Y|g'(\boldsymbol{X}, \boldsymbol{X}_c)) \neq P_2(Y|g'(\boldsymbol{X}, \boldsymbol{X}_c))$.

The risk over any distribution $P$ can be written as (using the cross-entropy loss),

$$\begin{aligned}
R_P(g) &= \mathbb{E}_P[\ell(Y, g'(\boldsymbol{X}, \boldsymbol{X}_c) h(\boldsymbol{X}_c))] \\
&= -\mathbb{E}_P[\sum_y y \log g'(\boldsymbol{X}, \boldsymbol{X}_c) h(\boldsymbol{X}_c)] \\
&= -\mathbb{E}_P[\sum_y y \log g'(\boldsymbol{X}, \boldsymbol{X}_c)] - \mathbb{E}_P[\sum_y y \log h(\boldsymbol{X}_c)]
\end{aligned} \tag{4}$$

The risk difference is,

$$\begin{aligned}
&R_{P_2}(g) - R_{P_1}(g) \\
&= \mathbb{E}_{P_1}[\sum_y y \log g'(\boldsymbol{X}, \boldsymbol{X}_c)] - \mathbb{E}_{P_2}[\sum_y y \log g'(\boldsymbol{X}, \boldsymbol{X}_c)] + \mathbb{E}_{P_1}[\sum_y y \log h(\boldsymbol{X}_c)] - \mathbb{E}_{P_2}[\sum_y y \log h(\boldsymbol{X}_c)] \\
&= \mathbb{E}_{P_1}[\sum_y y \log g'(\boldsymbol{X}, \boldsymbol{X}_c)] - \mathbb{E}_{P_2}[\sum_y y \log g'(\boldsymbol{X}, \boldsymbol{X}_c)]
\end{aligned}$$

---

[4] In practice, the constraint may be evaluated on an intermediate representation of $g$, such that $g$ can be written as, $g(\boldsymbol{X}) = g_1(\phi(\boldsymbol{X}))$ where $\phi$ denotes the representation function. However, for simplicity, we assume it is applied on $g(\boldsymbol{X})$.

where the third and fourth terms cancel out because $P_1(\boldsymbol{X}_c, Y) = P_2(\boldsymbol{X}_c, Y)$ and thus the risk of $h(\boldsymbol{X}_c)$ is the same across $P_1$ and $P_2$. However, the risk for $g'(\boldsymbol{X}, \boldsymbol{X}_c)$ is not the same since $P_1(Y|g'(\boldsymbol{X}, \boldsymbol{X}_c)) \neq P_2(Y|g'(\boldsymbol{X}, \boldsymbol{X}_c))$. Thus the absolute risk difference is non-zero,

$$|R_{P_2}(g) - R_{P_1}(g)| > 0 \tag{5}$$

and $g$ is not a risk-invariant predictor. Hence, satisfying conditional independencies that $\boldsymbol{X}_c$ satisfies is necessary for a risk-invariant predictor. □

**Remark.** In the above Theorem, we considered the case where $P(\boldsymbol{A}|Y)$ changes across distributions. In the case where $\boldsymbol{A}$ and $Y$ are independent, $P(\boldsymbol{A}|Y) = P(\boldsymbol{A})$ and thus $P(\boldsymbol{A})$ would change across distributions while $P(Y|\boldsymbol{A}) = P(Y)$ remained constant. Since $g'(\boldsymbol{X}, \boldsymbol{X}_c)$ depends on $\boldsymbol{A}$ and $\boldsymbol{X}_c$, we obtain $P_1(Y|g'(\boldsymbol{X}, \boldsymbol{X}_c)) = P_2(Y|g'(\boldsymbol{X}, \boldsymbol{X}_c))$. However, the risk difference can still be non-zero since $P_1(\boldsymbol{A}) \neq P_2(\boldsymbol{A})$ and the risk expectation $\mathbb{E}_P[\sum_y y \log g'(\boldsymbol{X}, \boldsymbol{X}_c)]$ is over $P(Y, \boldsymbol{X}_c, \boldsymbol{A})$.

## B.2 PROOF OF PROPOSITION 3.1

**Proposition 3.1.** *Given a causal DAG realized by specifying the target-attributes relationship in Figure 2a, the correct constraint depends on the relationship of label $Y$ with the attributes $\boldsymbol{A}$. As shown, $\boldsymbol{A}$ can be split into $\boldsymbol{A}_{\overline{ind}}$, $\boldsymbol{A}_{ind}$ and $E$, where $\boldsymbol{A}_{\overline{ind}}$ can be further split into subsets that have a causal ($\boldsymbol{A}_{cause}$), confounded ($\boldsymbol{A}_{conf}$), selected ($\boldsymbol{A}_{sel}$) relationship with $Y$ ($\boldsymbol{A}_{\overline{ind}} = \boldsymbol{A}_{cause} \cup \boldsymbol{A}_{conf} \cup \boldsymbol{A}_{sel}$). Then, the (conditional) independence constraints $\boldsymbol{X}_c$ should satisfy are,*

1. *Independent: $X_c \perp\!\!\!\perp A_{ind}$; $X_c \perp\!\!\!\perp E$; $X_c \perp\!\!\!\perp A_{ind}|Y$; $X_c \perp\!\!\!\perp A_{ind}|E$; $X_c \perp\!\!\!\perp A_{ind}|Y, E$*
2. *Causal: $X_c \perp\!\!\!\perp A_{cause}|Y$; $X_c \perp\!\!\!\perp E$; $X_c \perp\!\!\!\perp A_{cause}|Y, E$*
3. *Confounded: $X_c \perp\!\!\!\perp A_{conf}$; $X_c \perp\!\!\!\perp E$; $X_c \perp\!\!\!\perp A_{conf}|E$*
4. *Selected: $X_c \perp\!\!\!\perp A_{sel}|Y$; $X_c \perp\!\!\!\perp A_{sel}|Y, E$*

*Proof.* The proof follows from d-separation (Pearl, 2009) on the causal DAGs realized from Figure 2a. For each condition, *Independent*, *Causal*, *Confounded* and *Selected*, we provide the realized causal graphs below and derive the constraints.

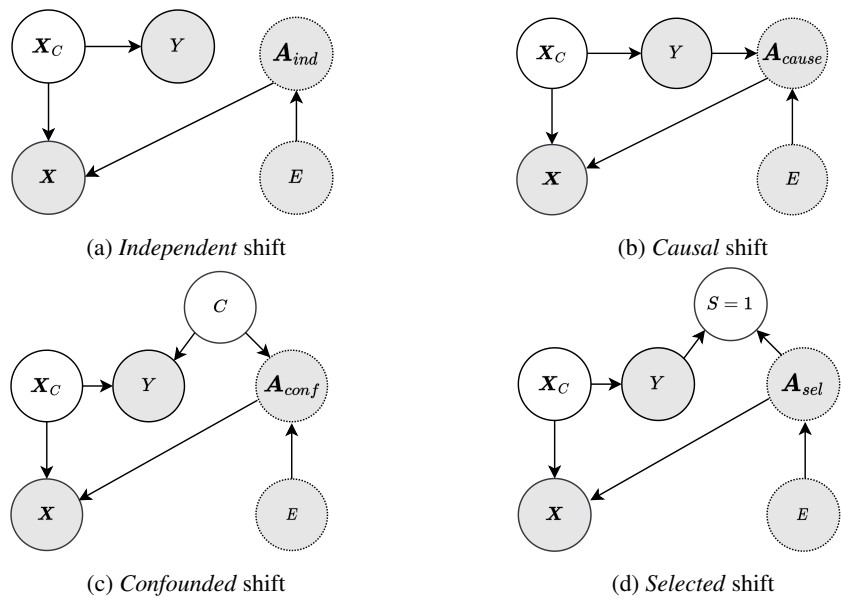

(a) *Independent* shift

(b) *Causal* shift

(c) *Confounded* shift

(d) *Selected* shift

Figure 3: Causal graphs for distinct distribution shifts based on $Y - \boldsymbol{A}$ relationship.

**Independent:** As we can see in Figure 3a, we have a collider $\boldsymbol{X}$ on the path from $\boldsymbol{X}_c$ to $\boldsymbol{A}_{ind}$ and $\boldsymbol{X}_c$ to $E$. Since there is a single path here, we obtain the independence constraints $\boldsymbol{X}_c \perp\!\!\!\perp \boldsymbol{A}_{ind}$ and $\boldsymbol{X}_c \perp\!\!\!\perp E$. Additionally, we see that conditioning on $Y$ or $E$ would not block the path from

$X_c$ to $A_{ind}$, which results in the remaining constraints: $X_c \perp\!\!\!\perp A_{ind} | Y$; $X_c \perp\!\!\!\perp A_{ind} | E$ and $X_c \perp\!\!\!\perp A_{ind} | Y, E$. Hence, we obtain,

$$X_c \perp\!\!\!\perp A_{ind} \, ; X_c \perp\!\!\!\perp E; X_c \perp\!\!\!\perp A_{ind} | Y; X_c \perp\!\!\!\perp A_{ind} | E; X_c \perp\!\!\!\perp A_{ind} | Y, E$$

**Causal:** From Figure 3b, we see that while the path $X_c \rightarrow X \rightarrow A_{cause}$ from $X_c$ to $A_{cause}$ contains a collider $X$, $X_c \not\perp\!\!\!\perp A_{cause}$ due to the presence of node $Y$ as a chain. By the d-separation criteria, $X_c$ and $A_{cause}$ are conditionally independent given $Y \implies X_c \perp\!\!\!\perp A_{cause} | Y$. Additionally, conditioning on $E$ is valid since $E$ does not appear as a collider on any paths between $X_c$ and $A_{cause} \implies X_c \perp\!\!\!\perp A_{cause} | Y, E$. We get the constraint $X_c \perp\!\!\!\perp E$ since all paths connecting $X_c$ to $E$ contain a collider (collider $X$ in $X_c \rightarrow X \rightarrow A_{cause} \rightarrow E$, collider $A_{cause}$ in $X_c \rightarrow Y \rightarrow A_{cause} \rightarrow E$). Hence, we obtain,

$$X_c \perp\!\!\!\perp A_{cause} | Y; X_c \perp\!\!\!\perp E; X_c \perp\!\!\!\perp A_{cause} | Y, E$$

**Confounded:** From Figure 3c, we see that all paths connecting $X_c$ and $A_{conf}$ contain a collider (collider $X$ in $X_c \rightarrow X \rightarrow A_{conf}$, collider $Y$ in $X_c \rightarrow Y \rightarrow C \rightarrow A_{conf}$). Hence, $X_c \perp\!\!\!\perp A_{conf}$. Additionally, conditioning on $E$ is valid since $E$ does not appear as a collider on any paths between $X_c$ and $A_{conf} \implies X_c \perp\!\!\!\perp A_{conf} | E$. We get the constraint $X_c \perp\!\!\!\perp E$ since all paths connecting $X_c$ and $E$ also contain a collider (collider $X$ in $X_c \rightarrow X \rightarrow A_{conf} \rightarrow E$, collider $Y$ in $X_c \rightarrow Y \rightarrow C \rightarrow A_{conf} \rightarrow E$). Hence, we obtain,

$$X_c \perp\!\!\!\perp A_{conf} \, ; X_c \perp\!\!\!\perp E; X_c \perp\!\!\!\perp A_{conf} | E$$

**Selected:** For the observed data, the selection variable is always conditioned on, with $S = 1$ indicating inclusion of sample in data. The selection variable $S$ is a collider in Figure 3d and we condition on it. Hence, $X_c \not\perp\!\!\!\perp A_{sel}$. Conditioning on $Y$ breaks the edge $X_c \rightarrow Y$, and hence all paths between $X_c$ and $A_{sel}$ now contain a collider (collider $X$ in $X_c \rightarrow X \rightarrow A_{sel}$) $\implies X_c \perp\!\!\!\perp A_{sel} | Y$. Additionally, conditioning on $E$ is valid since $E$ does not appear as a collider on any paths between $X_c$ and $A_{sel} \implies X_c \perp\!\!\!\perp A_{sel} | Y, E$. Hence, we obtain,

$$X_c \perp\!\!\!\perp A_{sel} | Y; X_c \perp\!\!\!\perp A_{sel} | Y, E$$

### B.2.1 PROOF OF COROLLARY 3.0.1

**Corollary 3.1.** *All the above derived constraints are valid for Graph 2a. However, in the presence of a correlation between $E$ and $X_c$ (Graph 2b), only the constraints conditioned on $E$ hold true.*

If there is a correlation between $X_c$ and $E$, $X_c \not\perp\!\!\!\perp E$. We can see from Figure 3 that in the presence of $X_c - E$ correlation, $X_c \not\perp\!\!\!\perp A_{ind}$; $X_c \not\perp\!\!\!\perp A_{ind} | Y$ (3a), $X_c \not\perp\!\!\!\perp A_{cause} | Y$ (3b), $X_c \not\perp\!\!\!\perp A_{conf}$ (3c) and $X_c \not\perp\!\!\!\perp A_{sel} | Y$ (3d). Hence, conditioning on environment $E$ is required for the valid independence constraints.

$\square$

### B.3 PROOF OF THEOREM 3.1

**Theorem 3.1.** *Under the canonical causal graph in Figure 2(a,b), there exists no (conditional) independence constraint over $\langle X_c, A, Y \rangle$ that is valid for all realized DAGs as the type of multi-attribute shifts vary. Hence, for any predictor algorithm for $Y$ that uses a single (conditional) independence constraint over its representation $\phi(X)$, $A$ and $Y$, there exists a realized DAG $\mathcal{G}$ and a corresponding training dataset such that the learned predictor cannot be a risk-invariant predictor for distributions in $\mathcal{P}_{\mathcal{G}}$, where $\mathcal{P}_{\mathcal{G}}$ is the set of distributions obtained by changing $P(A|Y)$.*

*Proof.* The proof follows from an application of Proposition 3.1 and Theorem 2.1.

**First claim.** Under the canonical graph from Figure 2(a or b), the four types of attribute shifts possible are *Independent*, *Causal*, *Confounded* and *Selected*. From the constraints provided for these four types of attribute shifts in Proposition 3.1, it is easy to observe that there is no single constraint that is satisfied across all four shifts. Thus, given a data distribution (and hence, dataset) with specific types of multi-attribute shifts such that $X_c$ satisfies a (conditional) independence constraint w.r.t.

a subset of attributes $\boldsymbol{A}_s \subseteq \boldsymbol{A}$, it is always possible to change the type of at least one of the those attributes' shifts to create a new data distribution (dataset) where the same constraint will not hold.

**Second Claim.** To prove the second claim, suppose that there exists a predictor for $Y$ based on a single conditional independence constraint over its representation, $\psi(\phi(\boldsymbol{X}), \boldsymbol{A}_s, Y)$ where $\boldsymbol{A}_s \subseteq \boldsymbol{A}$. Since the same constraint is not valid across all attribute shifts, we can always construct a realized graph $\mathcal{G}$ (and a corresponding data distribution) by changing the type of at least one attribute shift $A \in \boldsymbol{A}_s$, such that $\boldsymbol{X}_c$ would not satisfy the same constraint as $\phi(\boldsymbol{X})$. Further, under this $\mathcal{G}$, $\boldsymbol{X}_c$ would satisfy a different constraint on the same attributes. From Theorem 2.1, all conditional independence constraints satisfied by $\boldsymbol{X}_c$ under $\mathcal{G}$ are necessary to be satisfied for a risk-invariant predictor. Hence, for the class of distributions $\mathcal{P}_{\mathcal{G}}$, a single constraint-based predictor cannot be a risk-invariant predictor. □

**Corollary 3.2.** *Even when $|\boldsymbol{A}| = 1$, an algorithm using a single independence constraint over $\langle \phi(\boldsymbol{X}), A, Y \rangle$ cannot yield a risk-invariant predictor for all kinds of single-attribute shift datasets.*

*Proof.* Given a fixed (conditional) independence constraint over a predictor's representation, $\psi(\phi(\boldsymbol{X}), \boldsymbol{A}_s, Y)$, the proof of Theorem 3.1 relied on changing the target relationship type (and hence distribution shift type) for the attributes involved in the constraint. When $|A| = 1$, the constraint is on a single attribute $A$, $\psi(\phi(\boldsymbol{X}), A, Y)$ and the same proof logic follows. From Proposition 3.1, given a fixed constraint, we can always choose a single-attribute shift type (and realized DAG $\mathcal{G}$) such that the constraint is not valid for $\boldsymbol{X}_c$. Moreover, under $\mathcal{G}$, $\boldsymbol{X}_c$ would satisfy a different conditional independence constraint wrt the same attribute. From Theorem 2.1, since the predictor does not satisfy a conditional independence constraint satisfied by $\boldsymbol{X}_c$, it cannot be a risk-invariant predictor for datasets sampled from $\mathcal{P}_{\mathcal{G}}$. □

## C  *CACM* ALGORITHM

We provide the *CACM* algorithm for a general graph $\mathcal{G}$ below (Algorithm 1).

---
**Algorithm 1** *CACM*

---
**Input:** Dataset $(\boldsymbol{x}_i, \boldsymbol{a}_i, y_i)_{i=1}^n$, causal DAG $\mathcal{G}$
**Output:** Function $g(\boldsymbol{x}) = g_1(\phi(\boldsymbol{x})) : \boldsymbol{X} \to Y$
$\mathcal{A} \leftarrow$ set of observed variables in $\mathcal{G}$ except $Y, E$ (special domain attribute)
$C \leftarrow \{\}$ ▷ mapping of $A$ to $\boldsymbol{A}_s$
**Phase I:** Derive correct independence constraints
**for** $A \in \mathcal{A}$ **do**
   **if** $(\boldsymbol{X}_c, A)$ are d-separated **then**
      $\boldsymbol{X}_c \perp\!\!\!\perp A$ is a valid independence constraint
   **else if** $(\boldsymbol{X}_c, A)$ are d-separated conditioned on any subset $\boldsymbol{A}_s$ of the remaining observed variables in $\mathcal{A} \setminus \{A\} \cup \{Y\}$ **then**
      $\boldsymbol{X}_c \perp\!\!\!\perp A | \boldsymbol{A}_s$ is a valid independence constraint
      $C[A] = \boldsymbol{A}_s$
   **end if**
**end for**
**Phase II:** Apply regularization penalty using constraints derived
**for** $A \in \mathcal{A}$ **do**
   **if** $\boldsymbol{X}_c \perp\!\!\!\perp A$ **then**
      $RegPenalty_A = \sum_{|E|} \sum_{i=1}^{|A|} \sum_{j>i} \text{MMD}(P(\phi(\boldsymbol{x})|A_i), P(\phi(\boldsymbol{x})|A_j))$
   **else if** $A$ is in $C$ **then**
      $\boldsymbol{A}_s = C[A]$
      $RegPenalty_A = \sum_{|E|} \sum_{a \in \boldsymbol{A}_s} \sum_{i=1}^{|A|} \sum_{j>i} \text{MMD}(P(\phi(\boldsymbol{x})|A_i, a), P(\phi(\boldsymbol{x})|A_j, a))$
   **end if**
**end for**
$RegPenalty = \sum_{A \in \mathcal{A}} \lambda_A RegPenalty_A$
$g_1, \phi = \arg\min_{g_1, \phi}; \quad \ell(g_1(\phi(\boldsymbol{x})), y) + RegPenalty$

---

**Remark.** If $E$ is observed, we always condition on $E$ because of Corollary 3.1.

For the special case of Figure 2, *CACM* uses the following regularization penalty ($RegPenalty$) for *Independent*, *Causal*, *Confounded* and *Selected* shifts,

$$RegPenalty_{\boldsymbol{A}_{ind}} = \sum_{i=1}^{|E|} \sum_{j>i} \mathrm{MMD}(P(\phi(\boldsymbol{x})|a_{i,ind}), P(\phi(\boldsymbol{x})|a_{j,ind}))$$

$$RegPenalty_{\boldsymbol{A}_{cause}} = \sum_{|E|} \sum_{y \in Y} \sum_{i=1}^{|\boldsymbol{A}_{cause}|} \sum_{j>i} \mathrm{MMD}(P(\phi(\boldsymbol{x})|a_{i,cause}, y), P(\phi(\boldsymbol{x})|a_{j,cause}, y))$$

$$RegPenalty_{\boldsymbol{A}_{conf}} = \sum_{|E|} \sum_{i=1}^{|\boldsymbol{A}_{conf}|} \sum_{j>i} \mathrm{MMD}(P(\phi(\boldsymbol{x})|a_{i,conf}), P(\phi(\boldsymbol{x})|a_{j,conf}))$$

$$RegPenalty_{\boldsymbol{A}_{sel}} = \sum_{|E|} \sum_{y \in Y} \sum_{i=1}^{|\boldsymbol{A}_{sel}|} \sum_{j>i} \mathrm{MMD}(P(\phi(\boldsymbol{x})|a_{i,sel}, y), P(\phi(\boldsymbol{x})|a_{j,sel}, y))$$

# D  EXPERIMENTAL DETAILS

## D.1  DATASETS

**MNIST.**  Rotated (Ghifary et al., 2015) and Colored MNIST (Arjovsky et al., 2019) present distinct distribution shifts. While Rotated MNIST only has $\boldsymbol{A}_{ind}$ wrt. $rotation$ attribute ($R$), Colored MNIST only has $\boldsymbol{A}_{cause}$ wrt. $color$ attribute ($C$). We combine these datasets to obtain a multi-attribute dataset with $\boldsymbol{A}_{cause} = \{C\}$ and $\boldsymbol{A}_{ind} = \{R\}$. Each domain $E_i$ has a specific rotation angle $r_i$ and a specific correlation $corr_i$ between color $C$ and label $Y$. Our setup consists of 3 domains: $E_1, E_2 \in \mathcal{E}_{tr}$ (training), $E_3 \in \mathcal{E}_{te}$ (test). We define $corr_i = P(Y = 1|C = 1) = P(Y = 0|C = 0)$ in $E_i$. In our setup, $r_1 = 15°, r_2 = 60°, r_3 = 90°$ and $corr_1 = 0.9, corr_2 = 0.8, corr_3 = 0.1$. All environments have 25% label noise, as in (Arjovsky et al., 2019). For all experiments on MNIST, we use a two-layer perceptron consistent with previous works  (Arjovsky et al., 2019; Krueger et al., 2021).

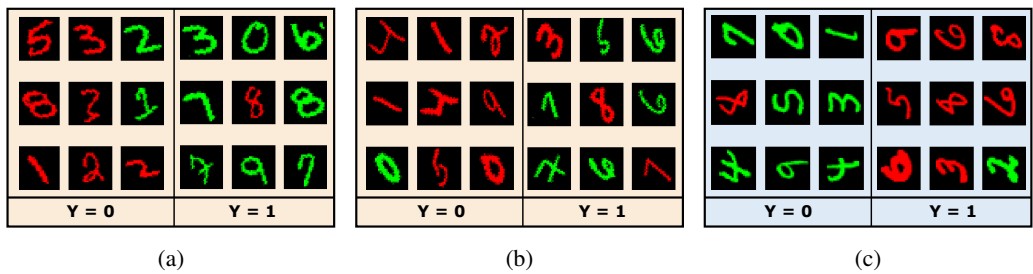

(a)        (b)        (c)

Figure 4: (a), (b) Train and (c) Test domains for MNIST.

**small NORB.**  Moving beyond simple binary classification, we use small NORB (LeCun et al., 2004), an object recognition dataset, to create a challenging setup with multi-valued classes and attributes over realistic 3D objects. It consists of images of toys of five categories with varying lighting, elevation, and azimuths. The objective is to classify unseen samples of the five categories. (Wiles et al., 2022) introduced single-attribute shifts for this dataset. We combine the *Causal* shift, $\boldsymbol{A}_{cause} = lighting$ wherein there is a correlation between lighting condition $lighting_i$ and toy

category $y_i$; and *Independent* shift, $\boldsymbol{A}_{ind} = azimuth$ that varies independently across domains, to generate our multi-attribute dataset $light + azi$. Training domains have 0.9 and 0.95 spurious correlation with $lighting$ whereas there is no correlation in test domain. We add 5% label noise in all environments. We use ResNet-18 (pre-trained on ImageNet) for all settings and fine tune for our task.

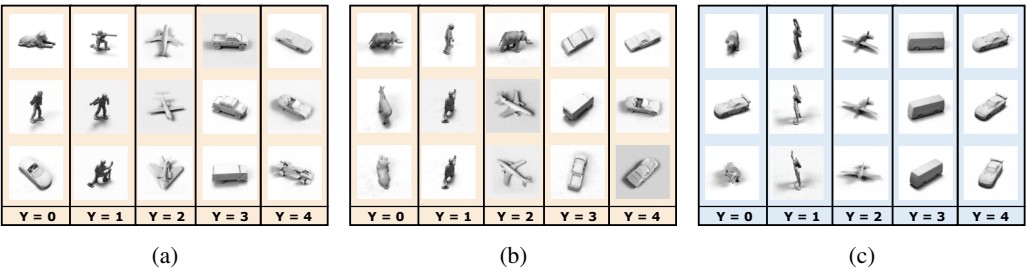

Figure 5: (a), (b) Train and (c) Test domains for small NORB.

**Waterbirds.** We use the Waterbirds dataset from Sagawa et al. (2020). This dataset classifies birds as "waterbird" or "landbird", where bird type ($Y$) is spuriously correlated with background ($bgd$) – "waterbird" images are spuriously correlated with "water" backgrounds (*ocean, natural lake*) and "landbird" images with "land" backgrounds (*bamboo forest, broadleaf forest*). Since background is selected based on $Y$, $\boldsymbol{A}_{cause} = background$. The dataset is created by pasting bird images from CUB dataset (Wah et al., 2011) onto backgrounds from the Places dataset (Zhou et al., 2018). There is 0.95 correlation between the bird type and background during training i.e., 95% of all waterbirds are placed against a water background, while 95% of all landbirds are placed against a land background. We create training domains based on background ($|\mathcal{E}_{tr}| = |\boldsymbol{A}_{cause}| = 2$) as in Yao et al. (2022). We evaluate using worst-group error consistent with past work, where a group is defined as ($background, y$). We generate the dataset using the official code from Sagawa et al. (2020) and use the same train-validation-test splits.

To create the multi-attribute shift variant of Waterbirds, we add weather effects ($\boldsymbol{A}_{ind}$) using the Automold library[5]. We add darkness effect (darkness coefficient = 0.7) during training with 0.5 probability and rain effect (rain type = 'drizzle', slant = 20) with 1.0 probability during test. Hence, $|\boldsymbol{A}_{ind}| = 3$ ({no effect, darkness, rain}). Weather effect is applied independent of class label $Y$. Our training domains are based on background and we perform worst-group evaluation, same as the setup described above. Examples from train and test domains for multi-attribute shift dataset are provided in Figure 6.

We use ResNet-50 (pre-trained on ImageNet) for all settings consistent with past work (Sagawa et al., 2020; Yao et al., 2022). All models were evaluated at the best early stopping epoch (as measured by the validation set), again consistent with Sagawa et al. (2020).

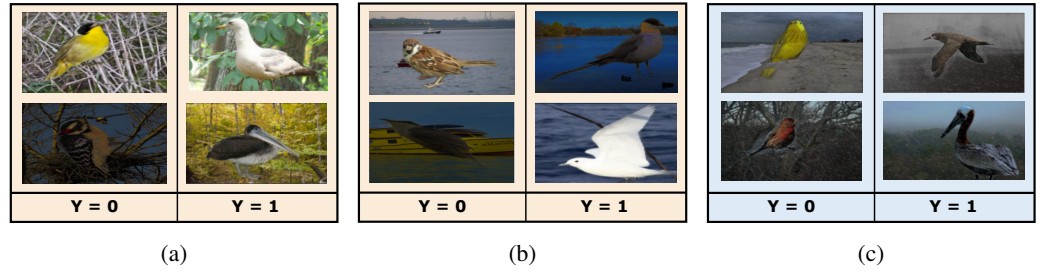

Figure 6: (a), (b) Train and (c) Test domains for Waterbirds.

---

[5] https://github.com/UjjwalSaxena/Automold--Road-Augmentation-Library

### D.2 Implementation details

All methods are trained using Adam optimizer. MNIST dataset is trained for 5000 steps (default in DomainBed (Gulrajani & Lopez-Paz, 2021)) while Waterbirds and small NORB are trained for 2000 steps. Consistent with the default value in DomainBed, we use a batch size of 64 per domain for MNIST. For small NORB, we use a batch size of 128 per domain and a batch size of 16 per domain for Waterbirds.

**Model Selection.** We create 90% and 10% splits from each domain to be used for training/evaluation and model selection (as needed) respectively. We use a validation set that follows the test domain distribution consistent with previous work on these datasets (Arjovsky et al., 2019; Ye et al., 2022; Wiles et al., 2022; Yao et al., 2022). Specifically, we adopt the *test-domain validation* from DomainBed for Synthetic, MNIST, and small NORB datasets where early stopping is not allowed and all models are trained for the same fixed number of steps to limit test domain access. For Waterbirds, we perform early stopping using the validation set consistent with past work (Sagawa et al., 2020; Yao et al., 2022).

**MMD implementation details.** We use the radial basis function (RBF) kernel to compute the MMD penalty. Our implementation is adopted from DomainBed (Gulrajani & Lopez-Paz, 2021). The kernel bandwidth is a hyperparameter and we perform a sweep over the hyperparamaeter search space to select the best RBF kernel bandwidth. The search space for hyperparameter sweeps is provided in Table 9, where $\gamma$ corresponds to 1/bandwidth.

***CACM* and baselines implementation details.** We provide the regularization constraints for different shifts used by *CACM* in Section C. For statistical efficiency, we use a single $\lambda$ value as hyperparameter for MNIST and small NORB datasets. The search space for hyperparameters is given in Table 9.

In MNIST and NORB, we input images and domains ($E = \boldsymbol{A}_{ind}$) to all baseline methods; *CACM* receives additional input $\boldsymbol{A}_{cause}$. Hence, in the *Independent* single-attribute shift, *CACM* and all baselines have access to exactly the same information. In Waterbirds, since $E$ is not defined in the original dataset, we follow the setup from Yao et al. (2022) to create domains based on backgrounds. Here, we provide images and domains ($E = background$) as input to all baselines except GroupDRO; to ensure fair comparison with GroupDRO, we follow Sagawa et al. (2020) and provide 4 *groups* as input based on $(background, y)$, along with images. For *CACM*, we do not use background domains but provide the attribute $\boldsymbol{A}_{cause} = background$ for the single-attribute dataset, and both $\boldsymbol{A}_{cause} = background$ and $\boldsymbol{A}_{ind} = weather$ for the multi-attribute shift dataset. Hence, for the single-shift Waterbirds dataset, all baselines receive the same information as *CACM*.

### D.3 Hyperparameter search

Following DomainBed (Gulrajani & Lopez-Paz, 2021), we perform a random search 20 times over the hyperparameter distribution and this process is repeated for total 3 seeds. The best models are obtained across the three seeds over which we compute the mean and standard error. The hyperparameter search space for all datasets and algorithms is given in Table 9.

## E Results

### E.1 Synthetic dataset

Our synthetic dataset is constructed based on the data-generating processes of the slab dataset (Mahajan et al., 2021; Shah et al., 2020). The original slab dataset was introduced by (Shah et al., 2020) to demonstrate the simiplicity bias in neural networks as they learn the linear feature which is easier to learn in comparison to the slab feature. Our extended slab dataset, adds to the setting from (Mahajan et al., 2021) by using non-binary attributes and class labels to create a more challenging task and allows us to study DG algorithms in the presence of linear spurious features.

Our dataset consists of 2-dimensional input $\boldsymbol{X}$ consisting of features $X_c$ and $A_{\overline{ind}}$. This is consistent with the graph in Figure 2 where attributes and causal features together determine observed features

$X$; we concatenate $X_c$ and $A_{\overline{ind}}$ to generate $X$ in our synthetic setup. Causal feature $X_c$ has a non-linear "slab" relationship with $Y$ while $A_{\overline{ind}}$ has a linear relationship with $Y$. We create three different datasets with *Causal* (E.1.1), *Confounded* (E.1.2) and *Selected* (E.1.3) $A_{\overline{ind}} - Y$ relationship respectively.

**Implementation details.** In all setups, $X_c$ is a single-dimensional variable and has a uniform distribution Uniform$[0, 1]$ across all environments. We use the default 3-layer MLP architecture from DomainBed and use mean difference (L2) instead of MMD as the regularization penalty given the simplicity of the data. We use a batch size of 128 for all datasets.

### E.1.1  *Causal* SHIFT

We have three environments, $E_1, E_2 \in \mathcal{E}_{tr}$ (training) and $E_3 \in \mathcal{E}_{te}$ (test). $X_c$ has a uniform distribution Uniform$[0, 1]$ across all environments.

$$y = \begin{cases} 0 & \text{if } X_c \in [0, 0.2) \\ 1 & \text{if } X_c \in [0.2, 0.4) \\ 2 & \text{if } X_c \in [0.4, 0.6) \\ 3 & \text{if } X_c \in [0.6, 0.8) \\ 4 & \text{if } X_c \in [0.8, 1.0] \end{cases}$$

$$A_{cause} = \begin{cases} y & \text{with prob.} = p \\ abs(y-1) & \text{with prob.} = 1-p \end{cases}$$

Hence, we have a five-way classification setup ($|Y| = 5$) with multi-valued attributes. Following (Mahajan et al., 2021), the two training domains have $p$ as 0.9 and 1.0, and the test domain has $p = 0.0$. We add 10% noise to $Y$ in all environments.

### E.1.2  *Confounded* SHIFT

We have three environments, $E_1, E_2 \in \mathcal{E}_{tr}$ (training) and $E_3 \in \mathcal{E}_{te}$ (test). $X_c$ has a uniform distribution Uniform$[0, 1]$ across all environments. Our confounding variable $c$ has different functional relationships with $Y$ and $A_{conf}$ which vary across environments.

$$c_{E_1, E_2} = \begin{cases} 1 & \text{with prob.} = 0.25 \\ 0 & \text{with prob.} = 0.75 \end{cases} \qquad c_{E_3} = \begin{cases} 1 & \text{with prob.} = 0.75 \\ 0 & \text{with prob.} = 0.25 \end{cases}$$

The true function for $Y$ is given by,

$$y_{true} = \begin{cases} 0 & \text{if } X_c \in [0, 0.25) \\ 1 & \text{if } X_c \in [0.25, 0.5) \\ 2 & \text{if } X_c \in [0.5, 0.75) \\ 3 & \text{if } X_c \in [0.75, 1.0] \end{cases}$$

Observed $Y$ and $A_{conf}$ are functions of confounding variable $c$ and their distribution changes across environments as described below:

$$y_{E_1, E_2} = \begin{cases} y_{true} + c & \text{with prob.} = 0.9 \\ y_{true} & \text{with prob.} = 0.1 \end{cases} \qquad y_{E_3} = y_{true}$$

$$A_{conf} = \begin{cases} 2 * c & \text{with prob.} = p \\ 0 & \text{with prob.} = 1-p \end{cases} \quad ; p_{E_1} = 1.0, p_{E_2} = 0.9, p_{E_3} = 0.8$$

### E.1.3 *Selected* SHIFT

*Selected* shifts arise due to selection effect in the data generating process and induce an association between $Y$ and $\boldsymbol{A}_{sel}$. A data point is included in the sample only if selection variable $S = 1$ holds; $S$ is a function of $Y$ and $\boldsymbol{A}_{sel}$. The selection criterion may differ between domains (Veitch et al., 2021).

We construct three environments, $E_1, E_2 \in \mathcal{E}_{tr}$ (training) and $E_3 \in \mathcal{E}_{te}$ (test). $X_c$ has a uniform distribution $\mathrm{Uniform}[0, 1]$ across all environments. Our selection variable $S$ is a function of $Y$ and $A_{sel}$. We add 10% noise to $Y$ in all environments.

$$X_c \sim \mathrm{Uniform}[0, 1]; \qquad A_{sel} \in \{1, 2, 3, 4\}$$

The true function for $Y$ is given by,

$$y_{true} = \begin{cases} 0 & \text{if } X_c \in [0, 0.25) \\ 1 & \text{if } X_c \in [0.25, 0.5) \\ 2 & \text{if } X_c \in [0.5, 0.75) \\ 3 & \text{if } X_c \in [0.75, 1.0] \end{cases}$$

The function used to decide the selection variable $S$ (and hence the selection shift) varies across environments through the parameter $p$.

$$S = 1 \quad if \begin{cases} A_{sel} + y = 4 & \text{with prob. } = \quad p \\ A_{sel} - y = 1 & \text{with prob. } = \quad 1 - p \end{cases} \quad ; p_{E_1} = 0.9, p_{E_2} = 1.0, p_{E_3} = 0.0$$

### E.2 A FIXED CONDITIONAL INDEPENDENCE CONSTRAINT CANNOT WORK ACROSS ALL SHIFTS

Here, we compare the performance of two popular independence constraints in the literature Mahajan et al. (2021): unconditional $\boldsymbol{X}_c \perp\!\!\!\perp \boldsymbol{A}|E$, and conditional on label $\boldsymbol{X}_c \perp\!\!\!\perp \boldsymbol{A}|Y, E$ (we condition on E for fully generality) on Synthetic *Causal*, *Confounded* and *Selected* shift datasets (Table 6).

We train a model using ERM (cross-entropy) where the representation is regularized using either of the constraints. As predicted by Theorem 3.1, neither constraint obtains best accuracy on all three datasets. The conditional constraint is better on $\boldsymbol{A}_{cause}$ and $\boldsymbol{A}_{sel}$ datasets, whereas the unconditional constraint is better on $\boldsymbol{A}_{conf}$, consistent with Proposition 3.1. Predictors with the correct constraint are also more risk-invariant, having lower gap between train and test accuracy.

Table 6: Comparison of constraints $\boldsymbol{X}_c \perp\!\!\!\perp \boldsymbol{A}|Y, E$ and $\boldsymbol{X}_c \perp\!\!\!\perp \boldsymbol{A}|E$ in *Causal*, *Confounded* and *Selected* shifts. $\boldsymbol{X}_c \perp\!\!\!\perp \boldsymbol{A}|Y, E$ is a correct constraint for *Causal* and *Selected* shift but invalid for *Confounded* shift; while $\boldsymbol{X}_c \perp\!\!\!\perp \boldsymbol{A}|E$ is correct for *Confounded* but invalid for *Causal*, *Selected*.

| Constraint | Accuracy | | | | | |
| | $\boldsymbol{A}_{cause}$ | | $\boldsymbol{A}_{conf}$ | | $\boldsymbol{A}_{sel}$ | |
| | train | test | train | test | train | test |
| --- | --- | --- | --- | --- | --- | --- |
| $\boldsymbol{X}_c \perp\!\!\!\perp \boldsymbol{A}|E$ | $96.5 \pm 0.2$ | $62.4 \pm 5.7$ | $81.1 \pm 2.0$ | $\mathbf{67.1 \pm 1.7}$ | $96.4 \pm 0.4$ | $72.3 \pm 0.9$ |
| $\boldsymbol{X}_c \perp\!\!\!\perp \boldsymbol{A}|Y, E$ | $89.1 \pm 3.8$ | $\mathbf{89.3 \pm 2.3}$ | $78.4 \pm 2.6$ | $60.3 \pm 1.2$ | $91.1 \pm 1.7$ | $\mathbf{88.7 \pm 0.9}$ |

### E.3 EFFECT OF VARYING REGULARIZATION PENALTY COEFFICIENT

To understand how incorrect constraints affect model generalization capabilities, we study the *Causal* shift setup in small NORB. From Theorem 3.1, we know the correct constraint for $\boldsymbol{A}_{cause}$: $\boldsymbol{X}_c \perp\!\!\!\perp \boldsymbol{A}_{cause}|Y, E$. In addition, we see the following invalid constraint, $\boldsymbol{X}_c \perp\!\!\!\perp \boldsymbol{A}_{cause}|E$. We compare the performance of these two conditional independence constraints while varying the regularization penalty coefficient ($\lambda$) (Figure 7). We perform our evaluation across three setups

with different spurious correlation values of training environments and have consistent findings. We observe that application of the incorrect constraint is sensitive to $\lambda$ (regularization weight) parameter : as $\lambda$ increases, accuracy drops to less than 40%. However, accuracy with the correct constraint stays invariant across different values of $\lambda$.

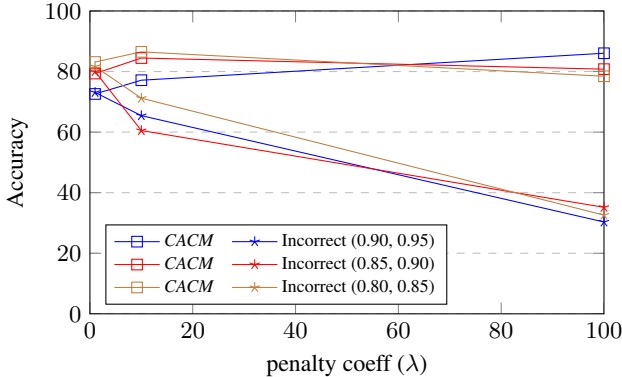

Figure 7: Accuracy of *CACM* ($\boldsymbol{X}_c \perp\!\!\!\perp \boldsymbol{A}_{cause} | Y, E$) and incorrect constraint ( $\boldsymbol{X}_c \perp\!\!\!\perp \boldsymbol{A}_{cause} | E$) on small NORB *Causal* shift with varying $\lambda$ {1, 10, 100} and spurious correlation in training environments (in parantheses in legend).

### E.4 PROVIDING ATTRIBUTE INFORMATION TO DG ALGORITHMS FOR A FAIRER COMPARISON

*CACM* leverages attribute labels to apply the correct independence constraints derived from the causal graph. However, existing DG algorithms only use the input features $\boldsymbol{X}$ and the domain attribute. Here we provide this attribute information to existing DG algorithms to create a more favorable setting for their application. We show that even in this setup, these algorithms are not able to close the performance gap with *CACM*, showing the importance of the causal information through graphs.

#### E.4.1 SYNTHETIC DATASET

We consider our Synthetic dataset with *Causal* distribution shift where our observed features $\boldsymbol{X} = (\boldsymbol{X}_c, \boldsymbol{A}_{cause})$. Note that by construction of $\boldsymbol{X}$, one of our input dimensions already consists of $\boldsymbol{A}_{cause}$. Hence, all baselines do receive information about $\boldsymbol{A}_{cause}$ in addition to the domain attribute $E$.

However, to provide a fairer comparison with *CACM*, we now *additionally* explicitly make $\boldsymbol{A}_{cause}$ available to all DG algorithms for applying their respective constraints by creating domains based on $\boldsymbol{A}_{cause}$ in our new setup. Using the same underlying data distribution, we group the data (i.e., create environments/domains) based on $\boldsymbol{A}_{cause}$ i.e, each environment $E$ has samples with same value of $\boldsymbol{A}_{cause}$.

In this setup (Table 7, third column), we see IB-IRM, DANN, and Mixup show significant improvement in accuracy but the best performance is still 14% lower than *CACM*. We additionally observe baselines to show higher estimate variance in this setup. This reinforces our motivation to use the causal graph of the data-generating process to derive the constraint, as the attribute values alone are not sufficient. We also see MMD, CORAL, GroupDRO, and MLDG perform much worse than earlier, highlighting the sensitivity of DG algorithms to domain definition. In contrast, *CACM* uses the causal graph to study the structural relationships and derive the regularization penalty, which remains the same in this new dataset too.

#### E.4.2 WATERBIRDS

We perform a similar analysis on the Waterbirds multi-attribute shift dataset. In order to provide the same information to other DG algorithms as *CACM*, we create domains based on $\boldsymbol{A}_{cause}$ x $\boldsymbol{A}_{ind}$ in this setup (Table 8). We observe mixed results – while some algorithms show significant improvement (ERM, IRM, VREx, MMD, MLDG, RSC), there is a performance drop for some others (IB-ERM, IB-IRM, CORAL, C-MMD, GroupDRO, Mixup). *CACM* uses the knowledge of the causal

Table 7: Synthetic dataset. Accuracy on unseen domain for *Causal* distribution shift when $A_{cause}$ is provided in input (column 2) and when $A_{cause}$ is additionally used to create domains (column 3).

| Algo. | Accuracy | |
|---|---|---|
| | $A_{cause}$ (input) | $A_{cause}$ (input+domains) |
| ERM | $73.3 \pm 1.3$ | $71.8 \pm 3.8$ |
| IB-ERM | $69.3 \pm 2.4$ | $68.2 \pm 4.9$ |
| IRM | $68.4 \pm 2.9$ | $64.1 \pm 0.8$ |
| IB-IRM | $67.8 \pm 2.6$ | $73.6 \pm 0.4$ |
| VREx | $77.4 \pm 1.2$ | $75.0 \pm 1.6$ |
| MMD | $72.3 \pm 4.3$ | $68.8 \pm 4.1$ |
| CORAL | $75.5 \pm 0.7$ | $72.1 \pm 0.8$ |
| DANN | $60.8 \pm 4.7$ | $65.8 \pm 11.9$ |
| C-MMD | $71.7 \pm 2.7$ | $67.9 \pm 4.9$ |
| CDANN | $71.1 \pm 2.5$ | $68.4 \pm 5.8$ |
| GroupDRO | $79.9 \pm 2.2$ | $65.4 \pm 3.4$ |
| Mixup | $58.3 \pm 1.8$ | $61.5 \pm 10.7$ |
| MLDG | $73.3 \pm 2.6$ | $65.3 \pm 3.3$ |
| SagNet | $72.5 \pm 2.3$ | $71.6 \pm 2.8$ |
| RSC | $70.9 \pm 3.4$ | $71.5 \pm 1.7$ |
| *CACM* | **$89.3 \pm 2.3$** | |

relationships between attributes and the label and hence the evaluation remains the same. Hence, we empirically demonstrate the importance of using information of the causal graph in addition to the attributes.

Table 8: Waterbirds. Accuracy on unseen domain for multi-attribute distribution shift when $A_{cause}$ is used to create domains (column 2) and when $A_{cause} \times A_{ind}$ is used to create domains (column 3).

| Algo. | Worst-group Accuracy | |
|---|---|---|
| | $A_{cause}$ | $A_{cause} \times A_{ind}$ |
| ERM | $37.0 \pm 1.1$ | $43.0 \pm 7.8$ |
| IB-ERM | $40.8 \pm 5.6$ | $34.4 \pm 1.0$ |
| IRM | $37.7 \pm 1.7$ | $42.2 \pm 2.5$ |
| IB-IRM | $46.9 \pm 6.5$ | $43.3 \pm 4.6$ |
| VREx | $38.1 \pm 2.3$ | $48.0 \pm 3.4$ |
| MMD | $45.2 \pm 2.4$ | $53.3 \pm 1.9$ |
| CORAL | $54.1 \pm 3.0$ | $47.5 \pm 2.8$ |
| DANN | $55.5 \pm 4.6$ | $57.7 \pm 6.5$ |
| C-MMD | $52.3 \pm 1.9$ | $45.9 \pm 4.9$ |
| CDANN | $49.7 \pm 3.9$ | $50.7 \pm 5.8$ |
| GroupDRO | $53.1 \pm 2.2$ | $40.9 \pm 3.1$ |
| Mixup | $64.7 \pm 2.4$ | $50.3 \pm 1.5$ |
| MLDG | $34.5 \pm 1.7$ | $43.6 \pm 3.8$ |
| SagNet | $40.6 \pm 7.1$ | $38.0 \pm 1.7$ |
| RSC | $40.9 \pm 3.6$ | $46.5 \pm 5.3$ |
| *CACM* | **$84.5 \pm 0.6$** | |

## F  $E - X_c$ RELATIONSHIP

The $E$-$X_c$ edge shown in Figure 2b represents correlation of $E$ with $X_c$, which can change across environments. As we saw for the $Y$-$A_{\overline{ind}}$ edge, this correlation can be due to causal relationship (Figure 8a), confounding with a common cause (Figure 8b), or selection (Figure 8c); all our results

(Proposition 3.1, Corollary 3.1) hold for any of these relationships. To see why, note that there is no collider introduced on $\boldsymbol{X}_c$ or $E$ in any of the above cases.

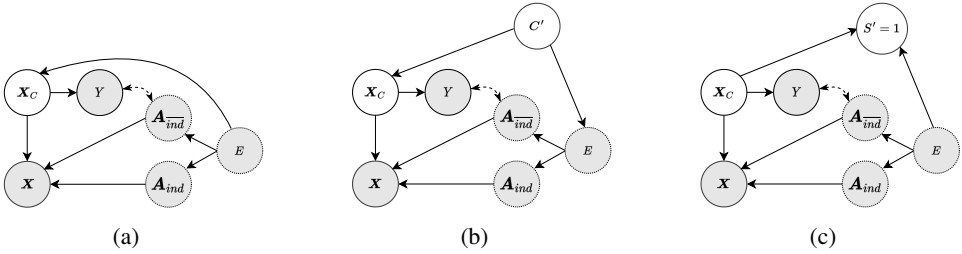

Figure 8: (a) Causal, (b) Confounded and (c) Selection mechanisms leading to $E$-$\boldsymbol{X}_c$ correlation.

## G ANTI-CAUSAL GRAPH

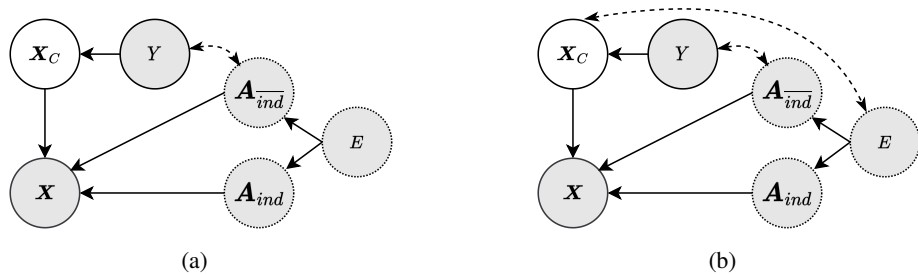

Figure 9: Corresponding anti-causal graphs for Figure 2. Note the graphs are identical to Figure 2 with the exception of the causal arrow pointing from $Y \longrightarrow \boldsymbol{X}_c$ instead of from $\boldsymbol{X}_c \longrightarrow Y$.

Figure 9 shows causal graphs used for specifying *multi-attribute* distribution shifts in an anti-causal setting. These graphs are identical to Figure 2, with the exception of change in direction of causal arrow from $\boldsymbol{X}_c \longrightarrow Y$ to $Y \longrightarrow \boldsymbol{X}_c$.

We derive the (conditional) independence constraints for the anti-causal DAG for *Independent*, *Causal*, *Confounded* and *Selected* shifts.

**Proposition G.1.** *Given a causal DAG realized from the canonical graph in Figure 9a, the correct constraint depends on the relationship of label $Y$ with the nuisance attributes $\boldsymbol{A}$. As shown, $\boldsymbol{A}$ can be split into $\boldsymbol{A}_{\overline{ind}}$, $\boldsymbol{A}_{ind}$ and $E$, where $\boldsymbol{A}_{\overline{ind}}$ can be further split into subsets that have a causal ($\boldsymbol{A}_{cause}$), confounded ($\boldsymbol{A}_{conf}$), selected ($\boldsymbol{A}_{sel}$) relationship with $Y$ ($\boldsymbol{A}_{\overline{ind}} = \boldsymbol{A}_{cause} \cup \boldsymbol{A}_{conf} \cup \boldsymbol{A}_{sel}$). Then, the (conditional) independence constraints that $\boldsymbol{X}_c$ should satisfy are,*

1. *Independent: $\boldsymbol{X}_c \perp\!\!\!\perp \boldsymbol{A}_{ind}$; $\boldsymbol{X}_c \perp\!\!\!\perp E$; $\boldsymbol{X}_c \perp\!\!\!\perp \boldsymbol{A}_{ind}\,|Y$; $\boldsymbol{X}_c \perp\!\!\!\perp \boldsymbol{A}_{ind}\,|E$; $\boldsymbol{X}_c \perp\!\!\!\perp \boldsymbol{A}_{ind}\,|Y, E$*
2. *Causal: $\boldsymbol{X}_c \perp\!\!\!\perp \boldsymbol{A}_{cause}\,|Y$; $\boldsymbol{X}_c \perp\!\!\!\perp E$; $\boldsymbol{X}_c \perp\!\!\!\perp \boldsymbol{A}_{cause}\,|Y, E$*
3. *Confounded: $\boldsymbol{X}_c \perp\!\!\!\perp \boldsymbol{A}_{conf}\,|Y$; $\boldsymbol{X}_c \perp\!\!\!\perp E$; $\boldsymbol{X}_c \perp\!\!\!\perp \boldsymbol{A}_{conf}\,|Y, E$*
4. *Selected: $\boldsymbol{X}_c \perp\!\!\!\perp \boldsymbol{A}_{sel}\,|Y$; $\boldsymbol{X}_c \perp\!\!\!\perp \boldsymbol{A}_{sel}\,|Y, E$*

*Proof.* The proof follows from *d*-separation using the same logic as earlier proof in Section B.2. We observe that for all attributes $A \in \boldsymbol{A}_{\overline{ind}}$ ($\boldsymbol{A}_{cause}$, $\boldsymbol{A}_{conf}$, $\boldsymbol{A}_{sel}$), it is required to condition on $Y$ to obtain valid constraints as $Y$ node appears as a chain or fork in the causal graph but never as a collider due to the $Y \longrightarrow \boldsymbol{X}_c$ causal arrow. $\qquad\square$

**Corollary G.1.** *All the above derived constraints are valid for Graph 9a. However, in the presence of a correlation between $E$ and $\boldsymbol{X}_c$ (Graph 9b), only the constraints conditioned on $E$ hold true.*

Table 9: Search space for random hyperparameter sweeps.

| Condition | Sweeps |
|---|---|
| MLP | learning rate: [1e-2, 1e-3, 1e-4, 1e-5] 
 dropout: 0 |
| ResNet | learning rate: [1e-2, 1e-3, 1e-4, 1e-5] 
 dropout: [0, 0.1, 0.5] |
| MNIST | weight decay: 0 
 generator weight decay: 0 |
| not MNIST | weight decay: $10^{\text{Uniform}(-6,-2)}$ 
 generator weight decay: $10^{\text{Uniform}(-6,-2)}$ |
| IRM | $\lambda$: [0.01, 0.1, 1, 10, 100] 
 iterations annealing: [10, 100, 1000] |
| IB-ERM, IB-IRM | $\lambda_{IB}$: [0.01, 0.1, 1, 10, 100] 
 iterations annealing$_{IB}$: [10, 100, 1000] 
 $\lambda_{IRM}$: [0.01, 0.1, 1, 10, 100] 
 iterations annealing$_{IRM}$: [10, 100, 1000] |
| VREx | $\lambda$: [0.01, 0.1, 1, 10, 100] 
 iterations annealing: [10, 100, 1000] |
| MMD | $\lambda$: [0.1, 1, 10, 100] 
 $\gamma$: [0.01, 0.0001, 0.000001] |
| CORAL | $\lambda$: [0.1, 1, 10, 100] |
| DANN, CDANN | generator learning rate: [1e-2, 1e-3, 1e-4, 1e-5] 
 discriminator learning rate: [1e-2, 1e-3, 1e-4, 1e-5] 
 discriminator weight decay: $10^{\text{Uniform}(-6,-2)}$ 
 $\lambda$: [0.1, 1, 10, 100] 
 discriminator steps: [1, 2, 4, 8] 
 gradient penalty: [0.01, 0.1, 1, 10] 
 adam $\beta_1$: [0, 0.5] |
| C-MMD | $\lambda$: [0.1, 1, 10, 100] 
 $\gamma$: [0.01, 0.0001, 0.000001] |
| GroupDRO | $\eta$: [0.001, 0.01, 0.1] |
| Mixup | $\alpha$: [0.1, 1.0, 10.0] |
| MLDG | $\beta$: [0.1, 1.0, 10.0] |
| SagNet | adversary weight: [0.01, 0.1, 1.0, 10.0] |
| RSC | feature drop percentage: $\text{Uniform}(0, 0.5)$ 
 batch drop percentage: $\text{Uniform}(0, 0.5)$ |
| *CACM* | $\lambda$: [0.1, 1, 10, 100] 
 $\gamma$: [0.01, 0.0001, 0.000001] |

