# OpenReview forum: "Modeling the Data-Generating Process is Necessary for Out-of-Distribution Generalization"
_ICLR.cc/2023/Conference — ICLR 2023 notable top 25%_

### Official Review · Reviewer_kwon · 2022-10-22

**Confidence:** 4
**Correctness:** 4
**Technical Novelty And Significance:** 3
**Empirical Novelty And Significance:** 3
**Recommendation:** 8

**Clarity, Quality, Novelty And Reproducibility:**

The manuscript is well written and easy to follow. Experiments appear to be of good quality.
Multi-attribute shifts

The level of reproducibility is on par with typical machine learning publications. Which is to say, not completely reproducible. Notably, the details behind the MMD regularization are missing. Which kernel was used to compute the MMD? What was the bandwidth? Was an approximation used for the kernel feature map?


**Strength And Weaknesses:**

This paper provides a framework for understanding domain generalization and when/why it sometimes fails. Further, the paper extends the typical domain generalization formulations to what the authors call multi-attribute shifts. Numerous baseline methods are used.

The authors mention the possibility that not all attributes are observed, yet give little information on the consequences of observing only some attributes. What are these consequences? What are the consequences or limitations of not observing E?
On a related note, the authors treat A a discrete. Does it have to be? Practically speaking, if A is discrete then it would seem that it must be of low cardinality as well (else one would need a lot of data).

By comparison, the NURD method appears to address unobserved attributes through the use of proxies (e.g. the border of an image) since it allows for high dimensional nuisance variables.
Further, the authors of NURD remark "Counterfactual invariance promises that a representation will not vary with the nuisance but it does not produce optimal models in general because it rejects models that depend on functions of the nuisance". They seem to argue that the uncorrelating property used in that work is superior as a result.

I would really like to see the authors compare to NURD, both in experiment and in discussion.
Puli, Aahlad Manas, et al. "Out-of-distribution Generalization in the Presence of Nuisance-Induced Spurious Correlations." International Conference on Learning Representations. 2021.

**Summary Of The Paper:**

The authors describe regularization based strategies for learning feature transformations that result in risk-invariant classifiers. They give two canonical generative DAGs and study domain generalization under these DAGs. The authors prove that any feature transformation must satisfy all relevant (conditional) independence relationships between the transformed features and the attributes and environment variables if it is risk-invariant (necessary condition). In particular, more than one independence relationship is required, which explains the inconsistent results from other domain generalization works. Finally, the authors propose an algorithm for identifying the set of required constraints when given a DAG or the shift-type of each attribute. The claims of the authors are demonstrated empirically through experimentations on three data sets.


**Summary Of The Review:**

A solid paper shedding light on domain generalization.

---

### Official Review · Reviewer_C2TU · 2022-10-23

**Confidence:** 3
**Correctness:** 4
**Technical Novelty And Significance:** 4
**Empirical Novelty And Significance:** 4
**Recommendation:** 8

**Clarity, Quality, Novelty And Reproducibility:**

The paper is very clear and very well-written. Several new ideas are proposed in the paper. For reproducibility, the authors provide the code for their experiments.

The authors provide a comprehensive discussion on how much use of incorrect constraints would hurt generalization.


**Strength And Weaknesses:**

Strengths:

This paper extends the view on the robustness toward distribution shift and brings the problem closer to the real-world scenario. The theoretical findings of this paper are not only important on their own but additionally demonstrate how the language of causal modeling could be used to correctly formulate the hypothesis about the independence constraints needed for method development. The authors provide a comprehensive discussion on how much the use of incorrect constraints would hurt generalization. The theoretical justifications in the paper are clear and convincing.

Weaknesses. A few things were not entirely clear to me:

- In the example of label shift due to more women coming to one of the hospitals: isn't it a case where the attribute "gender" is caudal to $X_c$? Isn't a case like this explicitly disallowed by the assumptions of the paper? Or is this case covered by a dotted arrow from E to $X_c$? From the description, it is not clear if the causation from the environment $E$ is also not allowed as causation from the attribute to the label? Also, in general, how restrictive is the assumption that attributes can't cause labels?
- The authors discuss that not all attributes have to be observed and that conditioning on the environment can substitute conditioning on the attribute. However, it is not clear to me, how it will hurt the performance of the method because this is a weaker constraint. I agree with the reasoning of the authors about the fact that in practical scenarios attribute shifts are sometimes known. However, even in the examples they describe, these distribution shifts are quite often discovered postfactum. Methods like IRM are designed to tackle the situation when the attributes are not explicitly observed. They don’t explicitly use the information about the attribute but tackle a more complicated problem when we have information that attributes shifts happened, but we don’t know in which way exactly. Intuitively, I don’t find it surprising that on the benchmarks introduced by the author or in general on any dataset which has explicit information about the attribute, the method which is using attribute is superior to the method that doesn’t. Would CACM still outperform IRM if only the environment was observed, but the attributes were not? Because for the case of IRM, it is not clear to me which constraint is violated.

**Summary Of The Paper:**

In this paper, the authors address the problem of a multi-attribute shift in Domain Generalization. They extensively and theoretically characterize various realizations of canonical causal graph modeling distribution shifts. They theoretically demonstrate that every such shift would entail a different independence constraint which explains previous empirical evidence that different domain generalization approaches that are focused only on a single attribute shift, show inconsistent performance between different datasets. The authors develop an algorithm (CACM) to tackle this problem of a multi-attribute shift in the case when the attributes are observed (or partially observed). The authors introduce 3 novel multi-attribute datasets, on which they demonstrate that CACM outperforms previously proposed benchmarks.


**Summary Of The Review:**

I think this is an important piece of work, which will have a direct contribution to the study of domain generalization in real-world scenarios, such as for example medical imaging domain. The paper is well-written, well-supported with theory and experiments, and contributes an important piece to the view of domain generalization in a practical setting.  For example, this paper is well aligned with the discussions on the understanding of causal modeling in medical imaging applications (Castro et al. 2020) and would facilitate further research into this direction in medical imaging with demonstrations on real-world medical datasets and benchmarks.

---

### Official Review · Reviewer_QsVS · 2022-10-24

**Confidence:** 4
**Correctness:** 3
**Technical Novelty And Significance:** 3
**Empirical Novelty And Significance:** 3
**Recommendation:** 6

**Clarity, Quality, Novelty And Reproducibility:**


This work is well written and easy to follow. Besides, I have the following concerns:


- Q: How is the DGPs, proposed methods and theory related to the work of combination shift of [Zhang et al. 2022]?


Zhang, Y., Wang, J., Xie, X., & Sugiyama, M. (2022). Equivariant Disentangled Transformation for Domain Generalization under Combination Shift. arXiv preprint arXiv:2208.02011.


- Q:  The algorithm CACM needs the correctly specified constraints, which might not be easy to achieve for some application scenarios. Would it be possible to automatically discover this kind of constraints?



- Q:  Regarding model selection, you used test-domain validation (oracle), as stated in Appendix D.2. Have you tried the training domain validation (from DomainBed)? Since model selection strategy would influence the results a lot and the training domain validation seems to be more realistic.




**Strength And Weaknesses:**


### Strength:

- The authors provide a new explanation from the perspective of data-generating process for the unexpected performance of DG algorithms.
- The causal modeling and analysis, along with the proposed solution are full of details, novel and interesting.
- The authors provide extensive empirical evidence to justify their claims and the effectiveness of CACM.



### Weaknesses

- The causal analysis seems to neglect the relationship between $A$ and $X_c$, which are discussed in IB-IRM [Ahuja et al. 2021]. Besides, the authors seem to miss a line of important related works, i.e., invariant graph learning, where [Chen et al. 2022] show that multiple potential distribution shifts can co-occur together in graphs and each of them can have a different causal relationship with the labels.

### References:

Ahuja, K., Caballero, E., Zhang, D., Gagnon-Audet, J. C., Bengio, Y., Mitliagkas, I., & Rish, I. (2021). Invariance principle meets information bottleneck for out-of-distribution generalization. Advances in Neural Information Processing Systems, 34, 3438-3450.

Yongqiang Chen et al., Invariance Principle Meets Out-of-Distribution Generalization on Graphs, ICML 2022: Workshop on Spurious Correlations, Invariance and Stability.



**Summary Of The Paper:**

This paper shows  that Modeling the Data-Generating Process is Necessary for Out-of-Distribution Generalization, otherwise any single, fixed constraint algorithms can fail under certain distribution shifts. The authors then propose Causally Adaptive Constraint Minimization (CACM) to adaptively identify and apply the correct independence constraints for regularization. The authors also provide extensive evidences to justify their claims both theoretically and empirically.

**Summary Of The Review:**

I enjoyed reading the paper. The authors identify an important issue in OOD generalization, provide detailed causal analysis as well as a novel solution.

---

### Official Review · Reviewer_LyJp · 2022-10-25

**Confidence:** 3
**Correctness:** 3
**Technical Novelty And Significance:** 3
**Empirical Novelty And Significance:** 3
**Recommendation:** 6

**Clarity, Quality, Novelty And Reproducibility:**

Clarity: Paper is well written. Most details are easy to understand. I mostly took a cursory look at the proofs and the main ones seem correct.

Quality: I believe the paper does a good job of motivating the need for the CACM algorithm.

Novelty: I believe that while the paper is sort of stating the obvious it is extremely crucial to make this point and I strongly believe there is value in unifying existing frameworks under a common canonical graph with a meta-algorithm that chooses a method based on the structure of the data.

Reproducibility: I did not run the code, but the authors have provided and brief look seems alright.

**Strength And Weaknesses:**


-> Not clear if Figure 2(c) is indeed exhaustive.

-> Second para on page 5 is insightful but the names are likely to create more confusion than they help. I urge authors to reconsider the naming.

-> Authors don't talk about whether imposing the independence constraints is *sufficient*.

-> Empirical evaluation: Ablation on results w/o regularization of the logits representation compared to benchmarks (E.3 is only evaluating for regularization penalty).

-> What conditional indpendence testing is used in empirical evaluation?

-> Can the authors specify how all hyperparameters are chosen? Is there a domain designated as a validation "domain"? There have been results showing inconclusivity of domain generalization purely due to hyperparameter tuning challenges.

-> When E and X_c are correlated, do use E conditioned regularization essentially reduces to regularizing only within domain? In that case the parameters/representations being shared is the more crucial contributor to the performance. May be I am missing something and the authors can clarify intuition.

-> Comment on generalizability of the canonical graph. It seems like the authors have come up with a canonical graph that encapsulates most of the methods and/or datasets. But it is not clear how exhaustive this graph is.

-> In the algorithm description, I found that you assume the test for independences are perfect. Is that reasonable?

**Summary Of The Paper:**

This work proposes to unify generalization methods by accounting for potential data-generating processes. The paper is presented to unify potential choice of generalization methods under a canonical causal graph which can account for i) label indepedent attributes, ii) label dependent attributes and iii) environment variables all of which can result in changes across distributions.

**Summary Of The Review:**

Overall I believe this is a valuable contribution. I would want a clarification on the above questions for me to more strongly endorse the paper and I look forward to the rebuttal from the authors.

---

### Decision · Program_Chairs · 2023-01-20

**Decision:**

Accept: notable-top-25%

**Justification For Why Not Higher Score:**

the proposed problem setting needs further studies to verify its practical value

**Justification For Why Not Lower Score:**

the paper is makes novel contributions and is of good technical quality

**Metareview: Summary, Strengths And Weaknesses:**

This paper revealed the fact that existing domain generalization methods do not work well on the multi-attribute shift datasets. To explain this phenomenon, this paper provides a formal characterization of generalization under multi-attribute shifts using a causal graph. Based on the insight, the authors further propose a method that makes use of the data generating process to improve the generalization. This paper has a new understanding of the ood problem and a novel method that is verified by extensive experiments. I would suggest acceptance of this paper given its technical novelty and good quality.


**Note From Pc:**

if the above contains the word "oral" or "spotlight" please see: "oral" presentation means -> notable-top-5% and "spotlight" means -> notable-top-25%. As stated in our emails, we are disassociating presentation type from AC recommendations